# Atomic emission detector with gas chromatographic separation and cryogenic pre-concentration (CryoTrap-GC-AED) for atmospheric trace gas measurements

Einar Karu[1], Mengze Li[1], Lisa Ernle[1], Carl A.M. Brenninkmeijer[1], Jos Lelieveld[1], Jonathan Williams[1]

[1]Atmospheric Chemistry Department, Max Planck Institute for Chemistry, 55128 Mainz, Germany

*Correspondence to*: Jonathan Williams (jonathan.williams@mpic.de)

**Abstract.** A gas detection system has been developed, characterized and deployed for pressurized gas phase sample analyses and near real-time online measurements. It consists of a cryogenic pre-concentrator (CryoTrap), a gas chromatograph (GC), and a new high-resolution atomic emission detector (AED III HR). Here the CryoTrap–GC–AED instrumental setup is presented and the performance for iodine ($1635 \pm 135$ counts I-atom$^{-1}$ pptv$^{-1}$), sulfur ($409 \pm 57$ counts S-atom$^{-1}$ pptv$^{-1}$), carbon ($636 \pm 69$ counts C-atom$^{-1}$ pptv$^{-1}$), bromine ($9.1 \pm 1.8$ counts Br-atom$^{-1}$ pptv$^{-1}$) and nitrogen ($28 \pm 2$ counts N-atom$^{-1}$ pptv$^{-1}$) emission lines is reported and discussed. The limits of detection (LODs) are in the low pptv range (0.5 – 9.7 pptv) and the signal is linear to at least 4 orders of magnitude, which makes it a suitable method for diverse volatile organic compound (VOC) measurements in the atmosphere, even in remote, unpolluted regions. The new system was utilized in a field study in a boreal forest at Hyytiälä, Finland in late summer 2016 which made monoterpene measurements possible among other VOCs. Furthermore, pressurized global whole-air sample, collected onboard the Lufthansa Airbus A340-600 IAGOS-CARIBIC aircraft in the upper troposphere and lower stratosphere region, were measured with the new setup providing data for many VOCs, including the long-lived organosulfur compound carbonyl sulfide.

## 1 Introduction

Atomic spectrometric analysis has been reported to provide highly sensitive detection a linear response of at least 5 orders of magnitude ($> 10^5$), and accurate elemental composition data of samples. If detection is preceded by analyte separation using gas chromatography (GC), compound specific data is attainable. The first atomic emission detector (AED) with a plasma as an excitation source and coupled with a GC was introduced in 1965 (McCormack et al., 1965). This group was the first to recognize the analytic potential of combining a GC separation with microwave induced plasma excitation and an electronic emission spectra detector. Early AEDs, using microwave induced plasmas, were operated at reduced pressures (Risby et al., 1983), until an atmospheric pressure version was developed in 1977 (Beenaker, 1977). The first commercial AED based on a microwave induced plasma and photodiode array detector coupled to a GC was released by Hewlett Packard in 1990 (Quimby and Sullivan, 1990). The first AED to measure atmospheric organosulfur species was reported in 1994 (Swan and Ivey, 1994). These systems also provided speciated element specific chromatograms and were used in a variety of applications including

the analysis of oils for sulfur containing compounds (Link et al., 2002). In contrast with the alternative analytical approach of gas chromatography coupled to mass spectrometry (GC–MS), the AED has the advantage of having highly selective wavelength dependant element specific detection and linear in detector response. Since elements are detected rather than molecules, potentially, the calibration of one carbon containing compound would allow calibration of all such compounds provided the response was indeed equimolar. That said it is important to keep in mind that a detector is often the last step in the whole analytical instrumental setup, therefore the response factor (RF) reflects the entire analyte pathway from the sample inlet up to the detector signal recording. This means that analyte losses by adsorption and absorption effects taking place in the transfer lines, pre-concentration stages, GC column and in the detector flow paths etc. contribute to the final RF. If such an equimolar response for the entire analytical system can be achieved, this would greatly simplify the calibration of complex mixtures and would even allow quantification of unidentified compound compounds not present in a calibration standard. For this reason, the AED has been used previously for the quantification of species that have been identified by GC-MS (Apel et al., 1998; Greenberg et al., 1999). Recently, further technical developments in the AED have led to improvements in sensitivity and furthermore, in the latest edition the whole AED III HR detector range is simultaneously measurable, making such systems even more attractive to atmospheric scientists. The simultaneous high-resolution wavelength recording capability of the new AED III detector (161 – 211 nm) in the CryoTrap–GC–AED system allows for screening for elements present in the speciated compounds, thus simplifying the identification of unknown compounds. In contrast, the AED II has a mechanically turning grating for measuring different wavelength groups in the range of 171 to 837 nm, however, due to the turning grating, simultaneous measurement of the whole range is not possible. The newest version of the AED detector system p potentially allows semiquantitative data on identified compounds in a chromatogram without a specific standard within the uncertainty range when a broad range of compound specific RFs are used for the determination of the single element RF. Such similar semiquantitative approaches can be also be used with other analytical techniques, like FID and MS, and this aspect of the AED performance will be assessed in this work

In this study, the instrumental setup and performance of a CryoTrap–GC–AED system, comprising of three commercially available units are examined. The performance for iodine (163 nm), sulfur (181 nm), carbon (193 nm), bromine (163 nm) and nitrogen (174 nm) emission lines is examined. The most sensitive atomic emission lines for the five elements in the range of the detector were chosen. The calibration linearities, limits of detection and compound specific response factors are reported for 64 compounds.

## 2 Experimental

The CryoTrap–GC–AED system consists of three stages: a liquid nitrogen based pre-concentration system (Entech model 7200, USA); a gas chromatographic separation (Agilent GC 7890B, USA); and a helium plasma based third generation atomic emission detector (Joint Analytical Systems AED III High Resolution, Germany). The schematic of the instrumental setup is shown in Fig. 1. Ultra-high purity helium (UHP, purity 99.9999%, Westfalen, Germany) flowing through a heated purifying

catalyst (Valco Instruments VICI, USA) is used throughout the system as the carrier and purging gas. For the reagent gases extra ultra-high purity $H_2$ (EUHP purity > 99.99999% by a Parker Balston Hydrogen Generator, model H2-300, Parker Hannifin Corporation, USA) and ultra-high purity $O_2$ (UHP, purity 99.9999%, Westfalen, Germany) were used. For the calibration gases and synthetic air as dilution gas Messer and Air Liquide gas regulators were used. For the certified ambient air calibration gas a pressure regulator completely made out of high-purity steel was used (Parker-Hannifin Veriflo 959, USA).

## 2.1 Cryogenic pre-concentration (CryoTrap)

The sample is introduced to the pre-concentration unit (CryoTrap) via an eight port multi position valve, consisting of the helium supply gas, four sample introduction inlets, an internal standard, a calibration standard and a blind port (Fig. 1, upper panel). The four sample introduction lines are each 2.0 m long (Restek Corp. Silcosteel, USA) with outer diameter 1/16″ (1.59 mm), inner diameter 0.040″ (1.02 mm). The sample is drawn onto the two enrichment traps via an evacuated volumetric reservoir, where the sample introduction volume is accurately determined by measuring the pressure at given temperature. All the CryoTrap internal flow path surfaces are coated with a thin high density ceramic Silonite-D layer to provide extremely inert surfaces, which nearly eliminates the adsorption of the analytes to the surfaces (Cardin, 1999).

The CryoTrap internal flow path is flushed with the sample gas before each pre-concentration step. After that the sample air is drawn (200 mL min$^{-1}$) through the first stage of pre-concentration, called the dehydration module (Fig. 1, middle panel), where $H_2O$ is selectively removed on an empty Silonite-D coated stainless-steel trap (outer diameter 1/8″ (3.18 mm), 31 cm long) held at -50°C. After collecting the required volume of sample, the trap is flushed with 75.0 mL of helium (100 mL min$^{-1}$) to remove any remaining air. Then module 1 is heated to 10°C and forward purged with 50.0 mL (10 mL min$^{-1}$) helium flow onto the main Tenax packed and Silonite-D coated volatile organic compound (VOC) enrichment trap (module 2, the second step of pre-concentration; outer diameter 1/8″ (1.02 mm), 31 cm long) held at -60°C. The forward helium purge helps to successfully transfer heavy, polar and semi-volatile organic compounds (SVOCs) onto the main module 2 VOC trap. The last pre-concentration step is pre-cooling of the cryo-focusing trap (module 3, unpacked ~4 cm long part of Silonite-D coated 1/32″ (0.79 mm) transfer line) to -180°C and thereafter kept at -160°C during module 2 back flushing at 230°C for 2.5 min, which will refocus the volatiles to a much smaller dead volume for splitless injection onto a GC column. The module 3 1/32″ (0.79 mm) transfer line is rapidly heated to 60°C for 2 min inside a sheathing 1/8″ (3.18 mm) perfluoroalkoxy alkane (PFA) tube with hot air supplied through the rotary plate kept at 100°C. This ensures rapid and splitless injection of analyte molecules to the GC column through a transfer line heated to 110°C. The pre-concentration unit is equipped with two bulkhead heaters in between the heated rotary plate and the traps for better water condensation management.

## 2.2 Gas chromatograph (GC)

Gas chromatography is a common choice for analytical separation of VOCs for measurement with various detectors (Bourtsoukidis et al., 2017; Apel et al., 1998). We use an Agilent 7890B GC for the compound separation. The GC was fitted with a Supelco SPB-624 capillary column (length × inner diameter: 60 m × 250 μm; film thickness 1.40 μm), which is an

intermediate polar, proprietary phase bonded fused silica GC column. The SPB-624 type columns are widely used for volatile

non-halogenated, halogenated, and aromatic compounds analysis.

Helium is used as the column carrier gas, of which the flow rate is controlled over the GC electronic pneumatic control (EPC) valve number 3. First, helium flows from the GC EPC valve into a 1/16″ (1.59 mm) stainless steel line connected to the CryoTrap rotary valve number 2 (Fig. 1). After the pre-concentration procedure the helium flow with the remobilized analytes is guided back to the GC oven through a heated (110°C) Silonite-D coated 1/32″ (0.79 mm) transferline. Then, in the GC oven

a Swagelok T-split union guides the flow to the analytical column for splitless injection. The other end of the GC capillary column was led directly to the AED cavity through a heated transferline (250°C).

## 2.3 Atomic emission detector, third generation (AED III)

The AED measures the energy emitted at characteristic wavelengths by sample atoms present in the helium (200 - 250 mL min$^{-1}$) plasma cavity to quantify their number in a chromatographic peak. Combining this data with GC analyte separation, the

amount of the substance can be quantitatively determined.

The helium carrier gas (3.5 mL min$^{-1}$) eluting from the GC is led to the AED cavity through the capillary column that is housed in the heated transfer line (250°C). The helium plasma discharge cavity is also kept at constant 250°C temperature. The mixture of the carrier (helium) and reagent gases hydrogen (at 12 psi (0.83 bar) supply pressure) and oxygen (15 psi (1.03 bar) supply pressure) flow through a fused silica discharge tube where the gases are ionized into a plasma state by microwave energy. In

the high-energy plasma the eluted sample compounds from the GC are broken down into free radicals, ions and atoms. As they return from their excited state to ground state configuration, light radiation is emitted in their element specific characteristic wavelengths. The emitted ultraviolet radiation passes through a fused silica lens and a narrow slit, and then is reflected onto a reflective holographic grating by a fixed mirror. The grating disperses the light into discrete vertical bandwidths along a plane-concave polychromatic grating. Thereafter, the grating reflects and focuses the light in the wavelength range 161 – 211 nm in

a horizontal plane onto two back-thinned charge-coupled devices (CCDs). Due to the physical gap between the two CCDs there is a 7 nm gap in the range of 183 – 190 nm of the spectra. The CCDs convert light intensities across the given spectrum into electrical currents which are recorded by the "AED III Instrument Control Software". The software automatically calibrates the received light intensity signal from the CCD diode numbers into wavelengths intensity data according to a calibration table. This process (called element installation in the software) takes place automatically before every sample

measurement. After each measurement, all the wavelength dependent emission data are stored with 0.01 nm resolution. An example of a 5 element simultaneously recorded chromatogram of the 84 component Apel-Riemer-2015 gas calibration standard is shown in Fig. 2.

## 2.4 Characterization experiments

All the characterization experiments started with a zero-air (hydrocarbon free synthetic air with 20.5% oxygen and nitrogen

rest (Westfalen, Germany), which flows through an extra catalyst kept at 500°C to remove the trace amounts of hydrocarbons)

measurement for determining the instrumental background. Furthermore, at the beginning of the experimental design the highest calibration standard MR level carryover and retention potentials were tested for all of the compounds of interest with the zero-air measurement directly after as the sample measurement. The trap back-flushing and bakeout times were adapted accordingly. With the 15 min bakeout step after the injection the modules 1 and 2 are heated to 150°C and 210°C respectively

and back-flushed with He flow (75 ml min$^{-1}$), whereas the bulkhead heaters are kept at 150°C.

Two independent gas calibration standards were used for the characterization experiments: the 84 component (each nominally at 50 ppbv mixing ratio (MR) in UHP nitrogen rest) Apel-Riemer-2015 gas calibration standard (Table S1) and the 30 component (each nominally at 4 ppbv MR in UHP nitrogen rest) non-methane hydrocarbon (NMHC) ozone precursor National Physical Laboratory 2017 (NPL-2017) primary calibration standard gas cylinders (Table S2).

The Apel-Riemer-2015 standard used for the characterization measurements was diluted with the same zero-air as described above with two mass flow controllers (MFC, EL-Flow, Bronkhorst, the Netherlands). The MFC for the calibration gas had a range up to 20 mL min$^{-1}$ and the MFC for zero-air a range up to 5 L min$^{-1}$. For these experiments a constant 1 L min$^{-1}$ zero-air flow was used and five steps (1, 5, 10, 15, 20 mL min$^{-1}$) of calibration gas were used, to produce nominal MRs of 50, 250, 500, 750 and 1 000 pptv. This MR range is relevant for many ambient VOC measurements. With each sample pre-concentration

a 1 400 ml aliquot of the diluted sample flow was trapped and the rest of the flow was led to lab exhaust. Calibration level measurements were repeated four times. No dilution for the NPL primary calibration standard was used and thus trapped directly in the amounts of 40, 60, 80 and 250 mL (producing MR in the range of 114, 171, 229 and 714 pptv for 1.4 L unknown sample relevance). The NPL calibration step measurements were also repeated four times.

With separate tests the maximum linearity range of the CryoTrap-GC-AED was tested with the undiluted Apel-Riemer-2015

calibration gas measurement (~ 50 ppbv per compound). For higher MR samples also a smaller sample volume can be trapped, making it possible to measure even higher MR samples. The system was tested to be linear to at least 4 orders of magnitude.

## 3 Instrument performance

The AED system was always recalibrated immediately prior to and during the measurement of the samples with unknown MRs, using the independent gas calibration standards: the Apel-Riemer-2015 and the NPL-2017 gas cylinders. In addition,

during the regular automated IAGOS-CARIBIC pressurized sample measurement procedure, the National Oceanic and Atmospheric Administration (NOAA) ambient air calibration standard and the NMHC NPL standards were periodically measured after every five samples. From the calibration standard measurements, the linearities (R-squared), limits of detection (LOD), response factors per atom and measurement uncertainty were determined for all quantifiable compounds in the standard and listed in Table 1 and 2.

The LODs were calculated from the average height of the lowest calibration levels and the average height of the noise signals corresponding to the same chromatogram for each separable and measurable compound of the calibration standards. Three

times signal-to-noise ratio (by peak height) was maintained to define the LOD, representing the ~99.7% confidence interval. The LOD calculation is represented by Eq. (1):

$$LOD = 3 \times \frac{\overline{noise} \times \overline{MR}}{\overline{peak\ height}},$$ (1)

where $\overline{noise}$ is the average height of the noise signal, $\overline{MR}$ is the average mixing ratio of the lowest level calibration level and $\overline{peak\ height}$ is the average peak height of that calibration level of the corresponding compound.

The average per atom RFs were calculated separately for each species using the average area under the peak curve ($\overline{AUC}$), the corresponding average MRs ($\overline{MR}$) and the number of atoms (NOA) present in the compound of interest as show by Eq. (2):

$$\overline{RF} = \frac{\overline{AUC}}{\overline{MR} \times \text{NOA}}.$$ (2)

The RFs were calculated separately for all compounds and their individual calibration MR levels and thereafter all the individual RFs over the whole calibration range were averaged to a single RF per compound.

**3.1 Carbon emission line performance**

The performance of the carbon emission line (193 nm) derived from measurements of three calibration standard gases is listed in Table 1. For Apel-Riemer-2015, the derived LODs are in the range 0.8 – 9.7 pptv. Each compound's RF is the average of

18 independent data points in the whole 50 – 1 000 pptv calibration range. The RFs for these compounds lie between 343 ± 147 counts C-atom$^{-1}$ pptv$^{-1}$ (benzyl chloride) to 800 ± 291 counts C-atom$^{-1}$ pptv$^{-1}$ (isobutyl nitrate) and on average 587 ± 91 counts C-atom$^{-1}$ pptv$^{-1}$ (47 compound mean). For NPL-2017, the LODs are in a smaller range of 0.5 – 1.1 pptv (more similar compounds) and the RFs per carbon atom range from 581 ± 23 counts C-atom$^{-1}$ pptv$^{-1}$ (1,2,4-trimethylbenzene) to 913 ± 77 counts C-atom$^{-1}$ pptv$^{-1}$ (toluene). The measured 22 compound NPL-2017 gas standard average RF was 684 ± 46 counts C-

atom$^{-1}$ pptv$^{-1}$ (about 17% higher compared to Apel-Riemer-2015 measurements and with about half the standard deviation). The both experiments average per carbon atom RF was determined as 636 ± 69 counts C-atom$^{-1}$ pptv$^{-1}$.

The CryoTrap-GC-AED system sensitivity and precision are also dependent on the discharge tube, as the two experiments were conducted with different discharge tubes (as the old one broke). Nevertheless, sample gas was always directly determined against calibration standards with the same discharge tube. If a discharge tube breakage took place, a complete new calibration

was carried out. The same 10 compounds (isobutene, 1,3-butadiene, pentane, isoprene, hexane, benzene, toluene, ethylbenzene, 1,3,5-trimethylbenzene and 1,2,4-trimethylbenzene) that were measured from both calibration standards are compared in Fig. 3. Three compounds: pentane, ethylbenzene and especially toluene do not adhere to the 1σ confidence interval on the one to one RF comparison between these two experiments. The reason remains unknown as there are other similar compounds to these in the comparison which stick to the correlation. The blanks analysed before and after the samples for both calibration

standards did not indicate contamination of these compounds.

Like the RFs, also the compound specific LODs depend on the CryoTrap pre-concentration efficiency, compound transmission efficiency through the gas transfer lines, GC separation and carry through efficiency and AED helium plasma atomization and excitation efficiency. Throughout the whole instrument development process, best efforts were undertaken to use suitable transfer lines and gas union (Swagelok, USA) materials like Silcosteel (fused silica coated stainless steel), stainless steel, PFA and Teflon for minimizing analyte flow path losses, i.e. through wall losses (absorption and adsorption effects), condensation etc. Furthermore, high purity stainless steel pressure regulators were used for calibration standard gas bottles with analyte free degassing O-rings.

The compound specific per carbon atom RFs from the NPL-2017 NMHC calibration standard are summarized in Fig. 4. The combined CryoTrap-GC-AED setup demonstrated a spread of RFs determined from the carbon 193 nm emission line intensities, which were between around 550 to 900 counts pptv$^{-1}$ C-atom$^{-1}$ for the NPL-2017 calibration standard, depending on the compound. The response factors for the more volatile species 1,3-butadiene to isoprene lie within the range $585 \pm 30$ to $660 \pm 27$ counts C-atom$^{-1}$ pptv$^{-1}$. However, the lower volatility compounds comprising 2-methylpentane to o-xylene have higher and more variable RFs ranging from $684 \pm 54$ to $913 \pm 77$ counts C-atom$^{-1}$ pptv$^{-1}$, see Table 1 (b) for compound based specifics. A general increasing variability in carbon response was visible with increasing carbon atom number, which can be also seen in Fig. 4. The three components of the instrument were considered as to the cause of these observed effects. The cryofocus unit will more effectively trap the low volatile species and so is unlikely to be the root cause. That the higher volatility species RFs lie reasonably close together which could suggest that the AED detector is functioning correctly, although it remains a possibility that larger molecules are less efficiently than 100% converted to atoms in the plasma. This is just a hypothesis and has not been tested. Given the abrupt change in RF from isoprene to 2-methyl pentane we consider the most likely explanation to be that despite the inert short sample lines some internal surface effects are causing the variation. As previous AED studies have shown constant C3 – C8 per carbon response, it is rather an indication that the variability in RF in this study increases due to CryoTrap trapping or inlet system. Thus, we see that the CryoTrap-GC-AED detected response must be considered as a whole for the entire instrument where the trapping and transmission efficiency is compound specific and can directly influence the final detected signal.As the per atom response factors were determined to be compound specific, for the field samples presented later in this work all the analytes of interest were always calibrated and quantified against a calibration standard, which was pre-concentrated and analyzed with exactly the same parameters as the samples. Therefore, all the losses and other effects in the analytical system leading to the variation in the RFs were accounted for. The same applies also to the other elements measured. The average of the compound specific median RFs is $663 \pm 66$ counts pptv$^{-1}$ C-atom$^{-1}$. Therefore, within the uncertainty range the compound independent calibration could be applied. It is possible to lower the uncertainty if a similar compound group's average RF is used for the target species. Nevertheless, for higher accuracy a direct calibration method is recommended. Throughout this study the direct calibration method was used.

The discrepancy between the two calibration standard determined RFs seen in Table 1 and 2 could rise from the point that the compounds in the NPL-2017 calibration standard are at nominal 4 ppbv and could be pre-concentrated directly, but the Apel-

Riemer-2015 compounds are nominally at 50 ppbv, thus requiring extra dilution before the trapping. The dilution with zero-air could introduce further uncertainty, as well as other unknown factors (e.g. wall losses) may also cause this discrepancy for the RFs determined from the Apel-Riemer-2015 calibration standard. The comparison overview of these two standards can be seen again in Fig. 3.

### 3.2 Iodine, sulfur, bromine and nitrogen emission line performances

The Apel-Riemer-2015 gas phase calibration standard was used to evaluate the performance of other available and relevant atomic emission lines for atmospheric chemistry (Table 2).

Methyl iodide (iodine wavelength measured at 178 nm) has LOD of 0.7 pptv, and RF of $1635 \pm 135$ counts I-atom$^{-1}$ pptv$^{-1}$. The LOD of methyl iodide on iodine emission line was 9.6 times more sensitive compared to carbon emission line. Carbonyl sulfide (OCS) and carbon disulfide ($CS_2$) had LODs of 1.9 and 1.8 pptv, respectively. OCS had RF of $342 \pm 34$ counts S-atom$^{-1}$ pptv$^{-1}$, and $CS_2$ had RF of $476 \pm 79$ counts S-atom$^{-1}$ pptv$^{-1}$. The sulfur emission line at 181 nm was determined to be on average 3.1 times more sensitive compared to the carbon 193 nm emission line, calculated from OCS and $CS_2$ LOD values. The average sulfur response factor determined from the two latter compounds is $409 \pm 57$ counts S-atom$^{-1}$ pptv$^{-1}$. Bromodichloromethane ($9.9 \pm 1.5$ counts Br-atom$^{-1}$ pptv$^{-1}$), 1,2-dibromoethane ($9.0 \pm 1.8$ counts Br-atom$^{-1}$ pptv$^{-1}$) and bromoform ($8.3 \pm 2.0$ counts Br-atom$^{-1}$ pptv$^{-1}$) at bromine emission line at 163 nm had significantly higher LODs of 115.7, 61.9 and 64.2 pptv, respectively compared to iodine, sulfur and carbon emission lines due to significantly higher background. The average per bromine atom response factor was determined $9.1 \pm 1.8$ counts Br-atom$^{-1}$ pptv$^{-1}$. Nevertheless, bromine emission line provides great selectivity, where the high background removes the complex background signal. Nitrogen at 174 nm proved to be the least sensitive element for AED III detector. Nitrogen containing compounds: acetonitrile, acrylonitrile, isopropyl nitrate, propyl nitrate and isobutyl nitrate varied significantly with their LODs (see Table 2), despite all containing a single nitrogen atom. The different transmission efficiency of each of the five nitrogen compounds through the system's flow paths could cause such differences. The average per nitrogen atom response factor determined from the five nitrogen compounds was $28 \pm 2$ counts N-atom$^{-1}$ pptv$^{-1}$. The different element count scales are non-related to each other. All the measured element's emission wavelength spectra were used independently, where the most sensitive emission line was chosen for each compound of interest. For example, in the case of OCS molecule the sulfur emission line was used.

### 3.3 Advantages and disadvantages of the CryoTrap-GC-AED

Considering all the species measured with the CryoTrap-GC-AED, the system proved to be very sensitive detecting iodine, sulfur and carbon elements with RFs $1635 \pm 135$ counts pptv$^{-1}$ I-atom$^{-1}$, $409 \pm 57$ counts pptv$^{-1}$ S-atom$^{-1}$ and $636 \pm 69$ counts C-atom$^{-1}$ pptv$^{-1}$, respectively. The different elemental emission wavelength intensity count scales are not directly comparable due to different elemental background noise levels. From these three elements the lowest background noise level was measured for iodine, followed by sulfur and then by carbon. The element background noise level determines the LODs for these elements in the same increasing order, also seen in the Table 1 and 2.

The advantage of the AED based system is the highly selective element specific information obtainable from the emission wavelength data, which can be useful in helping to determine unknowns in a complex matrix.Generally, the CryoTrap-GC-AED system is in the same sensitivity range as an FID detector (Baker et al., 2010). The electron capture detector (ECD) remains more sensitive and sometimes no pre-concentration of the air sample is necessary (e.g. Schuck et al., 2009) however,

an ECD is limited only to the electron capturing compounds. The GC-MS systems generally have high sensitivity and good selectivity but require separate calibration gases for all species to be quantified since response factors vary considerably. Modern online mass spectrometry (MS) detectors, especially the proton transfer reaction time-of-flight mass spectrometry (PTR-TOF-MS) provide high sensitivity (sub pptv) and highly time resolved (< 1 s) measurement techniques without the need for pre-concentration, e.g. the new Tofwerk / Aerodyne Research Vocus PTR-TOF (Krechmer et al. 2018). However, without

pre-separation with gas chromatography the chemical identity of signals can be ambiguous.

The specific emission wavelength range of the high-resolution AED III enables the measurement capability to accurately detect at least 11 elements (e.g. bromine at atomic emission wavelength of 163 nm is not mentioned in the vendors list, but could be accurately measured): antimony, arsenic, carbon, germanium, iodine, nitrogen, oxygen (requires 5% methane in 95% nitrogen reactant gas), phosphorus, selenium, silicon and sulfur, of which a maximum of 8 can be simultaneously saved from the raw

data (JAS product description from webpage, 2020). This makes the first screening of an unknown gas phase sample for compounds with various elements much easier and faster compared to more complex mass spectra. The AED capability to measure oxygen is not suitable for trace atmospheric measurements due to the requirement of 10% methane reagent mixture. As this high concentration of methane produces a very high background signal. This capability is suitable for oxygen measurements in oil and gas industry.

For the work presented here, a pre-concentrating GC-MS could have served as the same or better alternative for monitoring targeted species present in a calibration gas. Nevertheless, the AED system has advantages for searching for exotic heteroatom molecules in the atmosphere, for the non-carbon elements which are in the detector wavelength range. For example, for the analysis of volcanic emissions. Unfortunately, no new exotic species (containing of iodine, bromine, nitrogen, silicon, mercury, selenium and arsenic) were revealed in any of the sampled environments to date, although these environments were remote

(upper troposphere / lower stratosphere) or distant from pollution sources (Boreal forest).

Some disadvantages of the current AED III detector are the short lifetime of the fused silica tube where the plasma is being maintained and the early software / electronic issues inhibiting the proper acquisition of the data. The helium plasma is sustained in a small quartz tube (47 mm long, 1.0 mm inner diameter, 1.25 mm outer diameter) with a polyimide coating (27 mm long) in the middle part of the outer surface. The tube is positioned in a water-cooled cavity, maintained at 80°C. The

lifetime of the discharge tube was around 5 to 30 days, sometimes up to 3-4 months, depending on the sample throughput and quality of the replacement of the fragile discharge tube. The discharge tube could break at any moment without a warning during a measurement sequence, leading to the loss of a sample.

The high consumption of helium to maintain the plasma is expensive and of an environmental concern as helium is escaping from our atmosphere to space. Furthermore, the large consumption of liquid nitrogen (around 2 – 3 L sample[-1]) for the sample

cryogenic pre-concentration makes it demanding on operator time, logistical field support, and somewhat expensive to operate. The use of liquid nitrogen makes it difficult to operate this instrument in remote areas. Nevertheless, the CryoTrap-GC-AED instrument was taken to a field measurement campaign at Finnish boreal forest in 2016 where among the other species it contributed to the quantification of monoterpene compounds, when accounting for the directly measured $NO_3$ radical reactivity in the boreal forest (Liebman et al., 2018) and will be presented in section 4.1.

## 290   4 CryoTrap-GC-AED case studies

The newly developed instrument has been deployed in near real-time field measurements and in lab based pressurized canister measurements. In the following we demonstrate two case studies, one based in the forest in Hyytiala, Finland and the other in the home laboratory in Mainz, Germany.

### 4.1 Finnish boreal forest field measurements at Hyytiälä site

Boreal forest (taiga) makes up around 33% of Earth's forest cover, making it the largest terrestrial biome in the world. Even in the epoch of the Anthropocene about ¾ of the boreal forest remains natural (Brandt et al., 2013). The field campaign took place in a boreal forest measurement site SMEAR II (Station for Measuring Forest Ecosystem-Atmosphere Relations II) at Hyytiälä, Finland in September 2016 (Hari and Kulmala, 2005). The site is situated approximately 50 km away from the first more densely populated location, thus anthropogenic influence is relatively low, particularly when the wind masses originate

from the north with low human activity and pollution.

The instrumentation was installed in an air-conditioned measurement container maintained at 25°C temperature. The CryoTrap-GC-AED system measured ambient air which was pulled from the center of a shared 8.5 m tall high-flow inlet (15 cm diameter, flowrate ~ 10 $m^3$ $min^{-1}$). From the 8.5 m inlet the air was drawn to the container through a 15 m long ½″ (1.27 cm diameter) PFA tubing at a flowrate of ~ 20 L $min^{-1}$ (transmission time 3.3 s). The inlet line was insulated and heated (10°C

above ambient) to avoid sunlight interactions and condensation. Membrane filters (polytetrafluoroethylene with pore size 5.0 µm, 47 mm diameter by Sartorius AG, Germany) were used to avoid particles entering the tubing at the connection from the high-flow inlet and replaced every 5 days. From the 15 m long sampling line inside the container the cryogenic sample trapping phase took place with a flowrate of 200 mL $min^{-1}$ and a total of 1 400 mL air was trapped for each sample. After the injection of a pre-concentrated sample from the last focusing trap to the GC, the traps were baked out and the pre-concentration of a

next sample started in parallel with the previous GC run. This enabled mean sample throughput time intervals of 1 h 22 min. The average liquid nitrogen consumption rate was 2.5 L $h^{-1}$. The CryoTrap and AED parameters presented in Section 2 and GC program listed in Table 3a were used.

The diel cycles of isoprene and 5 monoterpene species (α-pinene, Δ-3-carene, β-pinene, camphene, d-limonene and isoprene) are presented in Fig. 5. All monoterpene species show daytime minima and night-time maxima. This is unexpected as the

emissions of monoterpenes are primarily temperature dependent and ambient temperatures were higher by day (Tarvainen et al., 2005). Here the recurring night-time MR maxima can be explained with the shallow, nocturnal boundary layer, typically

accompanied by a temperature inversion in the lower troposphere, effectively trapping the ground-level emissions (Liebmann et al., 2018). The measured monoterpene species and isoprene on average accounted for about 70% of the directly measured NO$_3$ reactivity at night-time and about 40% during daytime, published in a separate research article (Liebmann et al., 2018).

The diel cycles of carbonyl sulfide and carbon disulfide are shown in Fig. 6. OCS showed daytime maxima between 9:00 and 15:00 UTC (12:00 and 18:00 local time), which is unexpected as daytime uptake by vegetation is generally regarded as the dominant sink for OCS (Sandoval-Soto et al., 2005). Again, the previously discussed shallow nocturnal boundary layer plays a role, as much less in-mixing of higher concentration OCS from the free troposphere takes place during night-time. Furthermore, it confirms the recently published finding at the same measurement location that OCS uptake is light independent

and controlled by stomatal opening, and therefore stomatal conductance and OCS uptake can continue during night-time under these conditions (Kooijmans et al., 2017). The night-time uptake of OCS by vegetation should be further characterized and parameterized for 3D global chemistry-transport model applications. CS$_2$ did not show any significant diel cycle and MRs significantly varied between 0.5 and 5 pptv.

## 4.2 IAGOS-CARIBIC whole-air sample measurement with CryoTrap-GC-AED

The IAGOS-CARIBIC (In-service Aircraft for a Global Observing System – Civil Aircraft for the Regular Investigation of the Atmosphere Based on an Instrument Container) (Brenninkmeijer et al., 2007) is a regular observation project with scientific instruments on board commercial aircraft (Lufthansa Airbus A340-600 in cooperation with Lufthansa AG) which carries out atmospheric monitoring on a global scale, especially in the upper troposphere / lower stratosphere (UT/LS, ~ 10 – 12 km) region. Since December 2015, a new CryoTrap–GC–AED system has been applied to measure whole-air pressurized gas

samples that were taken as part of this project. Many VOCs such as NMHCs (Baker et al., 2010), sulfur containing species, oxygenated and halogenated trace gases can be measured in the C3 – C14 range. These species are important to tropospheric ozone production, stratospheric ozone depletion and radiative forcing.

The whole-air samples (WAS) were pressurized at about 4 bar absolute pressure with two triggered retrospective air collectors (TRACs) each consisting of 14 specially manufactured glass vessels (2.74 L), and a high-resolution sampler (HIRES) which

consists of 88 stainless steel canisters (1.0 L). Every time after the flights took place and the samplers were delivered to the lab, the measurement procedure began with the initial pressure measurement of all the 116 individual samples.

CryoTrap-GC-AED was applied to measure the IAGOS-CARIBIC TRAC and HIRES whole-air samples. At the beginning of each measurement sequence, a 2.0 m long sample line was connected to a TRAC or HIRES sampler port, and a leak check is conducted by evacuating the connected sample line to ~3.0 psi (0.21 bar) absolute pressure with the CryoTrap instrument

pump. After the pressure stabilization in the line, a leak check was carried out for 3 min. If the pressure reading was less than ±0.5 psi (0.03 bar) different, which is within the precision range of the pressure gauge, the line is considered leak tight. The CryoTrap traps Module 1 and Module 2 (Fig. 1) are heated at 150ºC and 220ºC, respectively for 20 min for conditioning. The GC oven program is summarized in the Table 3 below. Again, the three independent gas phase calibration standards (Apel-Riemer-2015, NPL-2017, NOAA-2017) were used for calibrating the system (Table S1 and S2). A 1 400 mL aliquot of an

IAGOS-CARIBIC air sample was pre-concentrated. Exactly the same conditions were used for the calibration standards (except NPL standard where 50 mL were pre-concentrated) in order to have the same conditions and to thereby minimize the instrumental uncertainty. The IAGOS-CARIBIC WAS measurement sequence with the CryoTrap-GC-AED starts with a calibration. The measurement of zero-air (produced the same way as described in Section 2) is followed by the certified ambient air standard (NOAA-2017) measurement. The latter is followed by diluted Apel-Riemer-2015 gas phase standard

measurement (as described in Section 2). After that the continuous measurement of the WAS samples started. The NOAA-2017 certified ambient air reference calibration standard and the NPL 30 component ozone precursor NMHC reference calibration standard were measured after every five WAS sample measurement to compensate for instrumental sensitivity drift throughout the analysis.

An example dataset of OCS MRs measured in the UT/LS region from two flights (Munich to Los Angeles and Munich to

Shanghai in February 2019) are depicted in Fig. 7. The OCS data combined with the other VOC species from the same samples was used to determine the global atmospheric and lower stratospheric lifetime, troposphere to stratosphere flux of OCS and the stratospheric sink using the linear relationship between the long-lived species MRs (and their variability) to their known atmospheric lifetimes. This will be discussed in detail in a separate research article (article under review).

**5 Conclusions**

The newly developed CryoTrap-GC-AED analytical instrument enables the measurement of gas phase samples in minute concentrations (low pptv level) suitable for ambient air VOC species measurements in the troposphere and lower stratosphere. With this instrument it is possible to measure samples from canisters (e.g. IAGOS-CARIBIC whole-air samples) and also directly with circa 1-hour sampling frequency at a measurement station (e.g. Hyytiälä Finnish boreal forest measurement campaign in 2016) (Karu, 2019; Liebmann et al., 2018). The instrument proved to be sensitive and linear over more than 4

orders of magnitude ($> 10^4$) and provides accurate element specific information.. The RFs variability can accounted for with direct calibration methods, which lead to more accurate results as were found out in this study. Despite the RF variability known compounds present in a chromatogram and not present in the calibration standard can be still quantified with an accuracy of around 30-40%. Thus, for more accurate results direct calibration against certified gas phase calibration standards with exactly the same measurement conditions and volumes is recommended. Further tests should be done to investigate the

causes of variable responses of the compounds as an overall equimolar response would considerably improve the systems utility.. The possibility to measure 11 elements and 8 of them simultaneously might allow discovery of new atmospheric species of interest, e.g. containing selenium or silicon atoms, particularly in marine or volcanically influence environments. The instrument would become significantly more reliable if longer lasting helium discharge tubes were developed and the commercial software and electronics were improved. For further developments of the detector the possibility to measure a

wider spectral range would allow detection of even more elements simultaneously.

*Competing interests*. The authors declare that they have no conflict of interest.

*Author contribution*: EK and JW developed the idea. EK established the new measurement technique. EK, ML and LE carried out the experiments. CAMB developed the CARIBIC air sampling systems. EK wrote the manuscript with support from ML and JW. All authors discussed the results, commented and helped to improve the manuscript.

*Acknowledgements*. We are very grateful to Claus Koeppel (Max-Planck-Institute for Chemistry, Mainz) for IAGOS-CARIBIC TRACs / HIRES WAS sampling and for conducting all the pre and after flight ground tests for accurate sample collection. We would also like to thank Florian Obersteiner (Karlsruhe Institute of Technology, Karlsruhe) for providing the very helpful IAU_Chrom data analysis software.

*Financial support*. EK was supported by the Max Planck Graduate Center with the Johannes Gutenberg-Universität Mainz (MPGC).

The article processing charges for this open-access publication were covered by the Max Planck Society.

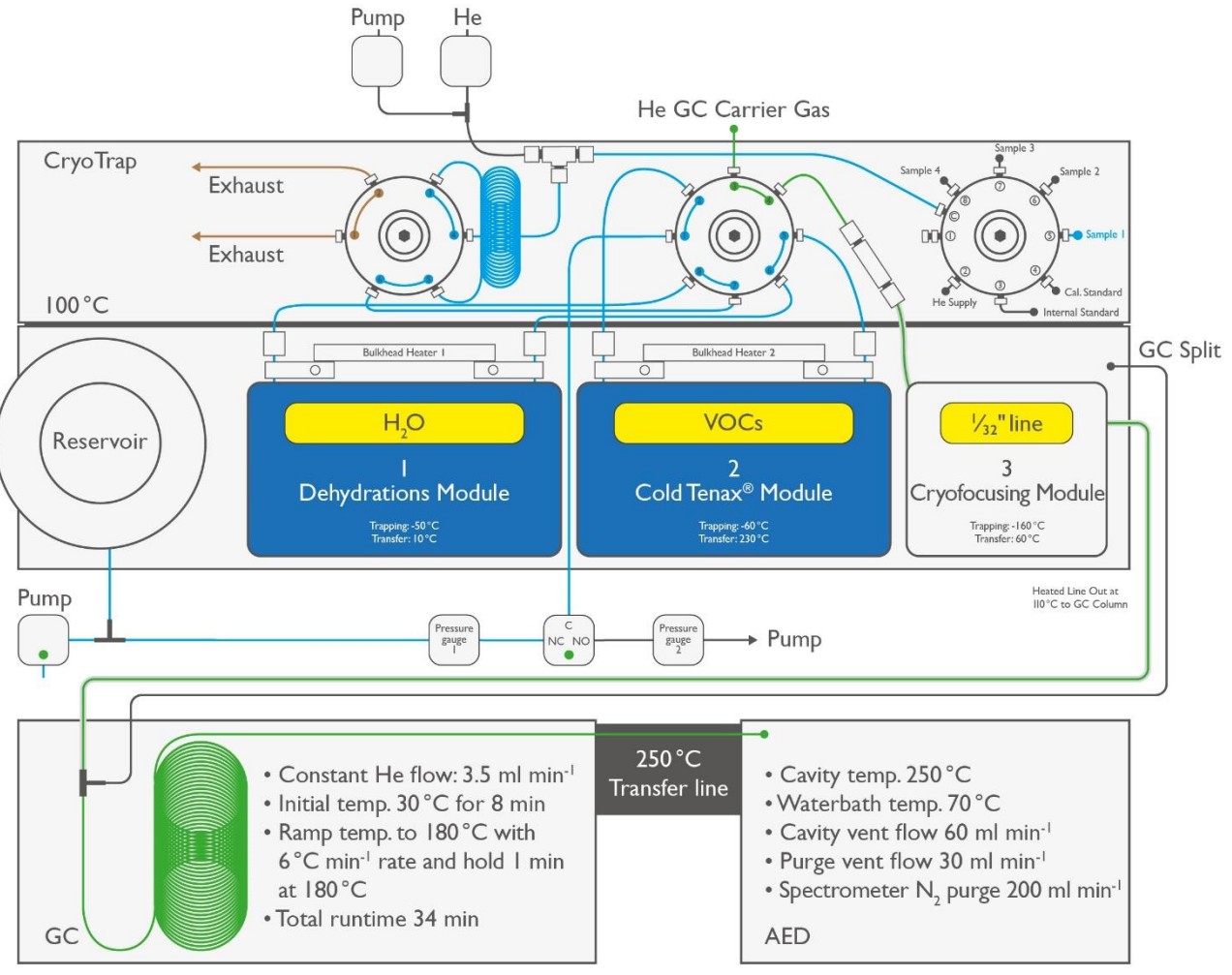

Figure 1: Schematic overview of instrumental setup, CryoTrap–GC–AED. Blue color indicates the CryoTrap sample pre-concentration flow pathway through tarp 1 and 2, starting from sample 1 inlet position. Green color indicates the GC-AED He flow through to the column and leading to the He plasma.

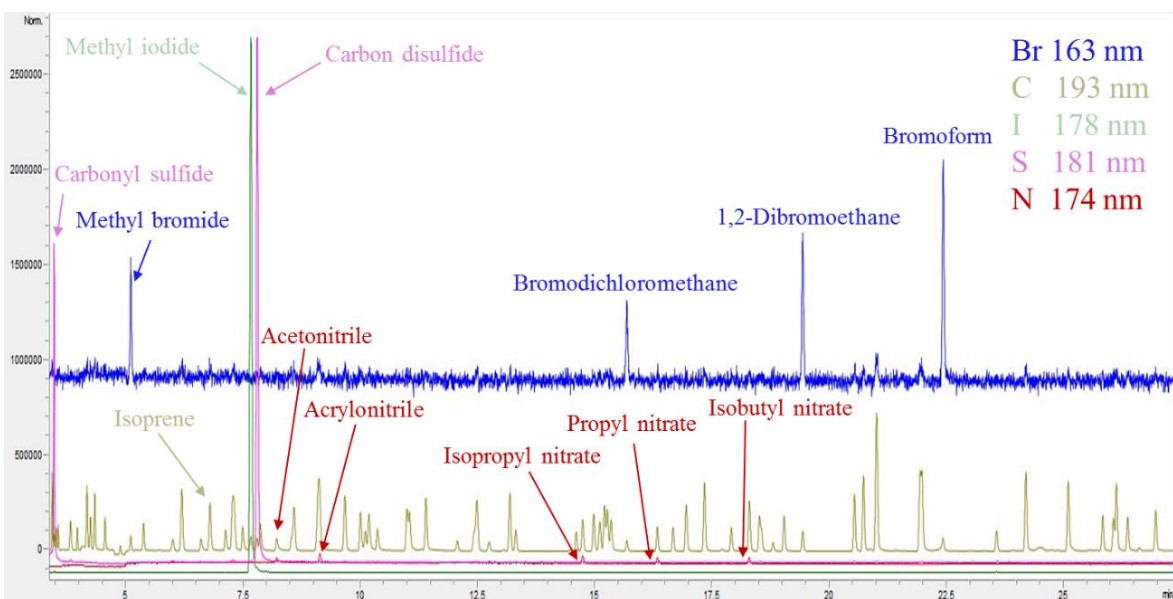

**Figure 2: Example CryoTrap-GC-AED normalized multi-element overlay chromatogram of ~250 pptv 84 component Apel-Riemer-2015 gas calibration standard.**

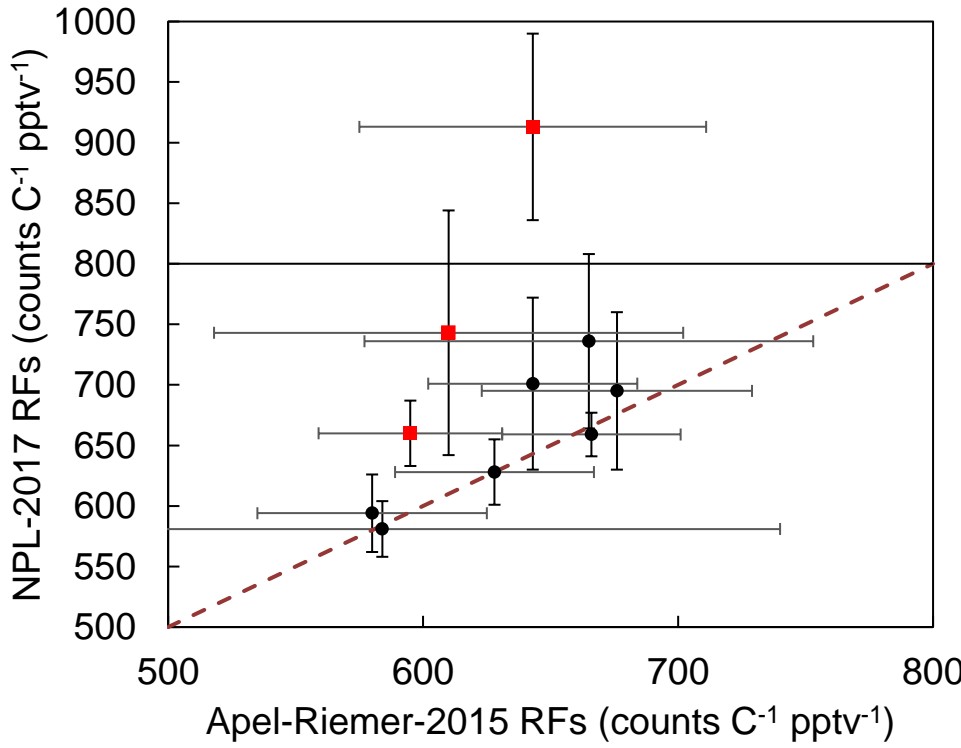

**Figure 3: Response factor comparison between the 10 common compounds measured from both Apel-Riemer-2015 and NPL-2017 gas calibration standards. The black circles show the average of all the calibration level data points for the according compounds measured in the Apel-Riemer-2015 and NPL-2017 gas calibration standards. The x and y-axis whiskers represent the standard**

 **deviation of all the measured data points for the according compound in the Apel-Riemer-2015 and NPL-2017 standards, respectively. The brown dashed line shows the one to one RF correlation between the two measurement experiments. The red squares with black whiskers show the pentane, ethylbenzene and toluene 1σ outliers (from bottom of the graph to top, respectively).**

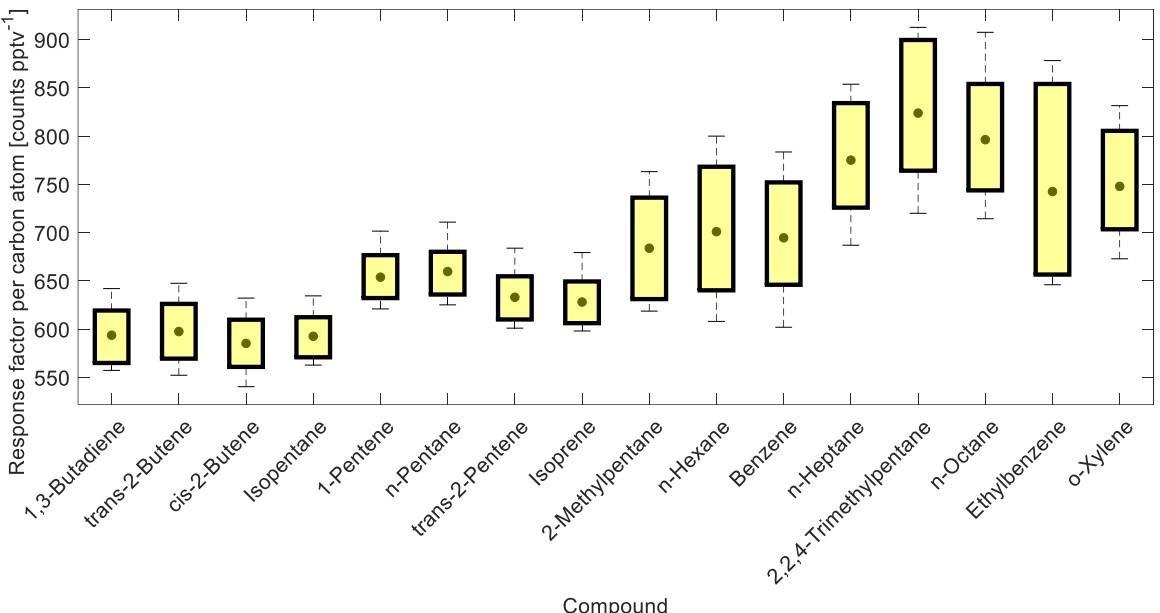

 **Figure 4: Boxplot graph from a selection of compound-specific per carbon atom RFs determined from the NPL-2017 NMHC primary calibration gas standard. The compound carbon emission line (193 nm) response factors as counts C-atom$^{-1}$ pptv$^{-1}$ are ordered by a growing number of carbon atoms on the x-axis. Central gray circles indicate the means, top and bottom edges of the box represent the 75$^{th}$ and 25$^{th}$ percentiles respectively, the whiskers show the single highest and lowest data point spread, N=176.**

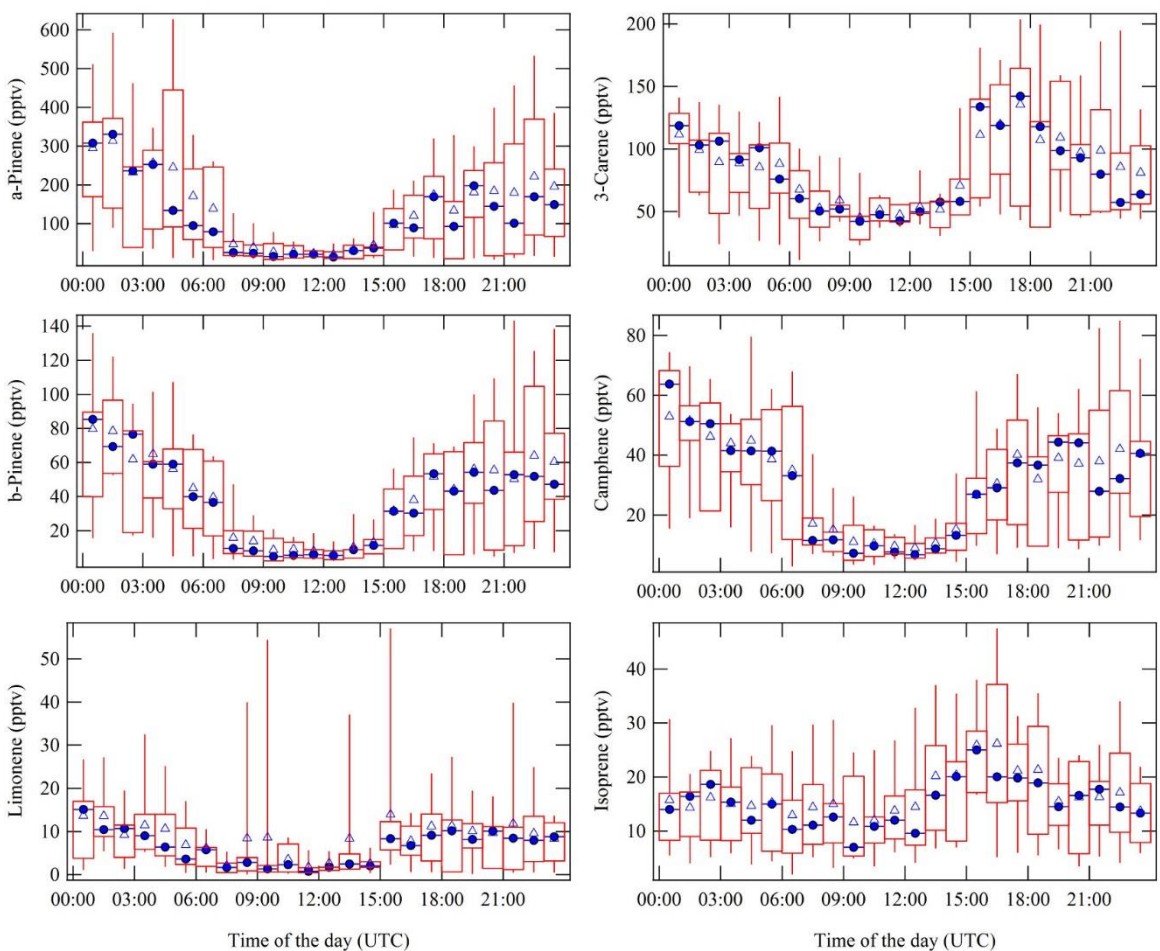

**Figure 5: Complete measurement period average diel cycles of α-pinene, Δ-3-carene, β-pinene, camphene, d-limonene and isoprene. For excluding the nearby sawmill (5 km in southeast direction) monoterpene emission transport event, the data from September 9 until 11 6:30 UTC were not included in the diel cycles. Blue triangles resemble the campaign overall hourly average and blue circles represent the hourly median values. The red rectangles with the whiskers show the data spread, where 50% of the data falls into the rectangles. The upper whiskers signify the upper 75-percentile data spread and the lower whiskers indicate the lower 25-percentile**
**of the data variation. Substantially higher MRs were observed at night-time compared to day.**

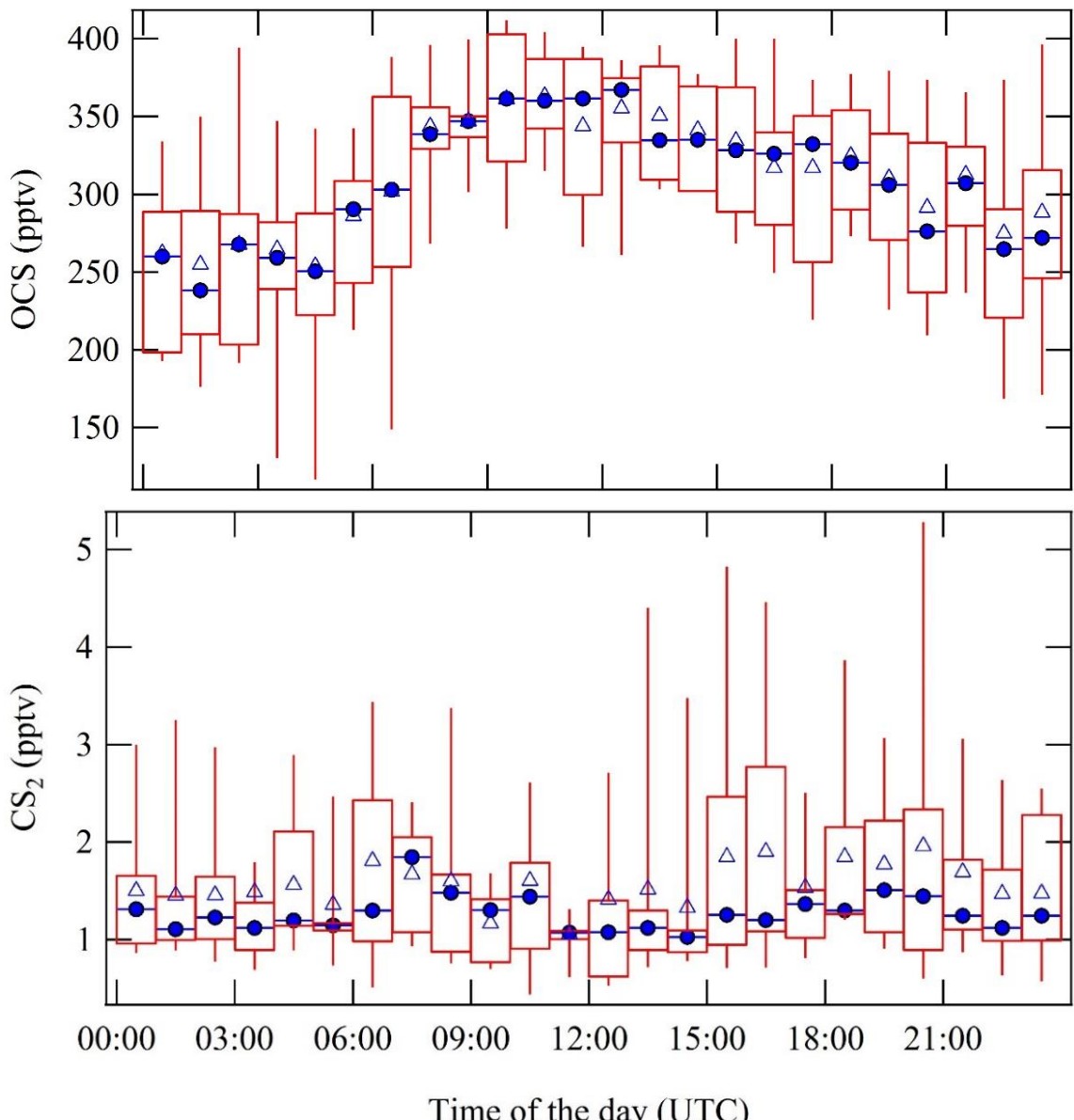

**Figure 6: Diel cycles of OCS MRs (upper) and CS₂ MRs (bottom) based on all AED measured data points throughout the measurement campaign. Blue triangles show the overall hourly mean and blue circles show the hourly median values. The red rectangles with the whiskers show the data spread, where 50% of the data falls into the rectangles. The upper whiskers represent the upper 75-percentile data spread and the lower whiskers denote the lower 25-percentile data variation.**

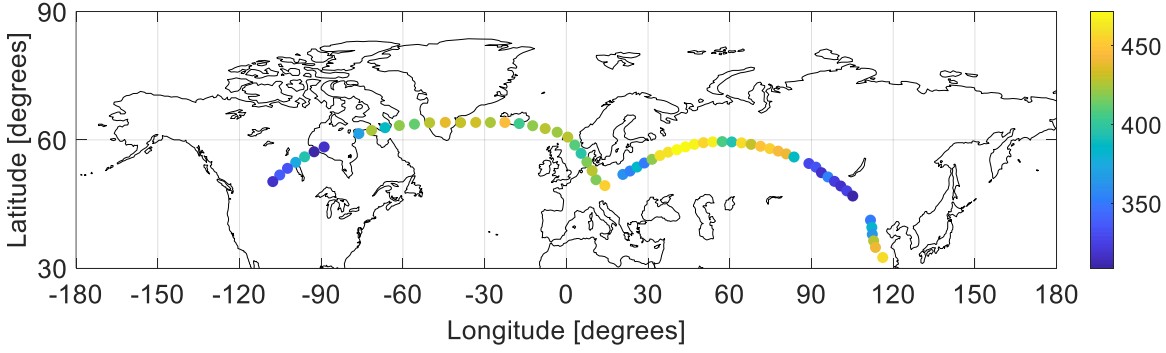

**Figure 7: Two IAGOS-CARIBIC example flights from Munich to Los Angeles and from Munich to Shanghai (in February 2019) with OCS MRs according to the color scale [pptv].**

Table 1: CryoTrap-GC-AED system limits of detection (LODs) on 3σ level, $R^2$ linearity and average response factors (RFs) per carbon atom values derived from (a) Apel-Riemer-2015 and (b) NPL-2017 gas calibration standard measurements ± standard deviation on 1σ level.

(a) Apel-Riemer-2015

| Compound | LOD (pptv) | $R^2$ | No. of C atoms | Average RF per C atom ± standard deviation (counts C-atom$^{-1}$ pptv$^{-1}$) |
|---|---|---|---|---|
| Carbonyl sulfide | 6.1 | 0.99832 | 1 | 637 ± 37 |
| Dichlorodifluoromethane (R-12) | 2.7 | 0.99189 | 1 | 589 ± 39 |
| Chlorodifluoromethane (HCFC-22) | 2.4 | 0.99589 | 1 | 745 ± 32 |
| Chloromethane | 2.9 | 0.99581 | 1 | 661 ± 43 |
| Methanol | 3.7 | 0.92247 | 1 | 529 ± 172 |
| Methyl bromide | 5.1 | 0.99429 | 1 | 553 ± 64 |
| Trichlorofluoromethane | 5.6 | 0.99431 | 1 | 652 ± 34 |
| Methyl iodide | 6.7 | 0.99668 | 1 | 552 ± 68 |
| Carbon disulfide | 5.4 | 0.99015 | 1 | 696 ± 109 |
| Chloroform | 5.7 | 0.99474 | 1 | 652 ± 55 |
| Tetrachloromethane | 7.0 | 0.99447 | 1 | 637 ± 69 |
| Bromoform | 5.8 | 0.99384 | 1 | 737 ± 57 |
| Vinyl chloride | 2.0 | 0.99492 | 2 | 549 ± 46 |
| Acetaldehyde | 1.8 | 0.99402 | 2 | 636 ± 35 |
| Chloroethane | 2.6 | 0.99469 | 2 | 550 ± 46 |
| Acetonitrile | 4.8 | 0.96653 | 2 | 523 ± 133 |

| | | | | |
|---|---|---|---|---|
| 1,1-Dichloroethane | 3.9 | 0.97800 | 2 | 430 ± 35 |
| 1,2-Dichloroethane | 3.3 | 0.99494 | 2 | 563 ± 46 |
| Trichloroethylene | 9.7 | 0.95093 | 2 | 457 ± 280 |
| 1,1,2-Trichloroethane | 3.3 | 0.99287 | 2 | 715 ± 99 |
| 1,2-Dibromoethane | 3.4 | 0.99489 | 2 | 638 ± 98 |
| 1,1,2,2-Tetrachloroethane | 3.2 | 0.99468 | 2 | 654 ± 100 |
| Propene | 0.8 | 0.99478 | 3 | 620 ± 33 |
| 2-Propanol | 1.7 | 0.99457 | 3 | 698 ± 37 |
| Isopropyl nitrate | 2.2 | 0.99886 | 3 | 742 ± 137 |
| Propyl nitrate | 9.6 | 0.99813 | 3 | 439 ± 232 |
| cis-1,3-Dichloropropene | 2.8 | 0.99452 | 3 | 512 ± 84 |
| Isobutene | 0.9 | 0.99532 | 4 | 666 ± 35 |
| 1,3-Butadiene | 1.0 | 0.99492 | 4 | 580 ± 45 |
| Methacrolein | 1.5 | 0.99458 | 4 | 728 ± 79 |
| Butanol | 2.5 | 0.99537 | 4 | 384 ± 42 |
| Isobutyl nitrate | 2.0 | 0.99878 | 4 | 800 ± 291 |
| Pentane | 1.1 | 0.98933 | 5 | 595 ± 36 |
| Isoprene | 1.2 | 0.99494 | 5 | 628 ± 39 |
| Hexane | 1.0 | 0.99496 | 6 | 643 ± 41 |
| Benzene | 0.9 | 0.99451 | 6 | 676 ± 53 |
| 4-Methyl-2-pentanone (MIBK) | 1.9 | 0.99513 | 6 | 566 ± 150 |
| 2-Hexanone | 2.7 | 0.99284 | 6 | 357 ± 129 |
| Hexanal | 6.1 | 0.98819 | 6 | 386 ± 182 |
| Chlorobenzene | 1.4 | 0.99540 | 6 | 573 ± 107 |
| 1,3-Dichlorobenzene | 2.8 | 0.98789 | 6 | 399 ± 133 |
| Benzyl chloride | 4.7 | 0.97921 | 6 | 343 ± 147 |
| 1,2-Dichlorobenzene | 2.6 | 0.99017 | 6 | 412 ± 125 |
| Toluene | 0.9 | 0.99502 | 7 | 643 ± 68 |
| Ethylbenzene | 0.9 | 0.99536 | 8 | 610 ± 92 |
| 1,3,5-Trimethylbenzene | 0.9 | 0.99470 | 9 | 665 ± 88 |
| 1,2,4-Trimethylbenzene | 1.1 | 0.99135 | 9 | 584 ± 156 |

(b) NPL-2017

| Compound | LOD (pptv) | $R^2$ | No. of C atoms | Average RF per C atom ± standard deviation (counts C-atom$^{-1}$ pptv$^{-1}$) |
|---|---|---|---|---|
| Propene & Propane | 0.8 | 0.99444 | 3 | 627 ± 18 |
| Isobutane | 0.8 | 0.99164 | 4 | 659 ± 18 |
| 1,3-Butadiene | 0.8 | 0.99432 | 4 | 594 ± 32 |
| trans-2-Butene | 0.8 | 0.99083 | 4 | 598 ± 33 |
| cis-2-Butene | 1.0 | 0.99129 | 4 | 585 ± 30 |
| Isopentane | 1.1 | 0.99598 | 5 | 593 ± 26 |
| 1-Pentene | 0.9 | 0.99263 | 5 | 654 ± 26 |
| n-Pentane | 0.9 | 0.99263 | 5 | 660 ± 27 |
| trans-2-Pentene | 0.9 | 0.99026 | 5 | 633 ± 27 |
| Isoprene | 1.0 | 0.99181 | 5 | 628 ± 27 |
| 2-Methylpentane | 0.9 | 0.97498 | 6 | 684 ± 54 |
| n-Hexane | 0.8 | 0.96687 | 6 | 701 ± 71 |
| Benzene | 0.7 | 0.96992 | 6 | 695 ± 65 |
| n-Heptane | 0.6 | 0.99386 | 7 | 775 ± 62 |
| Toluene | 0.5 | 0.99127 | 7 | 913 ± 77 |
| 2,2,4-Trimethylpentane | 0.6 | 0.99205 | 8 | 824 ± 76 |
| n-Octane | 0.5 | 0.98977 | 8 | 796 ± 68 |
| Ethylbenzene | 0.5 | 0.92709 | 8 | 743 ± 101 |
| o-Xylene | 0.5 | 0.99552 | 8 | 748 ± 60 |
| 1,3,5-Trimethylbenzene | 0.5 | 0.96353 | 9 | 736 ± 72 |
| 1,2,4-Trimethylbenzene | 0.6 | 0.99727 | 9 | 581 ± 23 |
| 1,2,3-Trimethylbenzene | 0.6 | 0.99681 | 9 | 619 ± 26 |

**Table 2: CryoTrap-GC-AED system limits of detection (LODs) on 3σ level, $R^2$ linearity and response factors (RFs) per iodine, sulfur, bromine and nitrogen atom values derived from Apel-Riemer-2015 calibration standard measurements ± standard deviation on 1σ level.**

| Compound | LOD (pptv) | $R^2$ | Element | Average RF per atom ± standard deviation (counts atom$^{-1}$ pptv$^{-1}$) |
|---|---|---|---|---|
| Methyl iodide (iodine) | 0.7 | 0.99668 | I | 1635 ± 135 |
| Carbonyl sulfide (sulfur) | 1.9 | 0.99964 | S | 342 ± 34 |

| | | | | |
|---|---|---|---|---|
| Carbon disulfide (sulfur) | 1.8 | 0.99966 | S | $476 \pm 79$ |
| Bromodichloromethane (bromine) | 115.7 | 0.98930 | Br | $9.9 \pm 1.5$ |
| 1,2-Dibromoethane (bromine) | 61.9 | 0.99226 | Br | $9.0 \pm 1.8$ |
| Bromoform (bromine) | 64.2 | 0.99082 | Br | $8.3 \pm 2.0$ |
| Acetonitrile (nitrogen) | 265.1 | 0.98281 | N | $19.8 \pm 2.7$ |
| Acrylonitrile (nitrogen) | 139.4 | 0.99657 | N | $28.2 \pm 0.8$ |
| Isopropyl nitrate (nitrogen) | 64.6 | 0.98954 | N | $35.8 \pm 2.9$ |
| Propyl nitrate (nitrogen) | 82.0 | 0.99180 | N | $25.8 \pm 2.2$ |
| Isobutyl nitrate (nitrogen) | 57.1 | 0.99060 | N | $28.2 \pm 2.1$ |

**Table 3: The GC oven programs for (a) Finish boreal forest and (b) IAGOS-CARIBIC measurements.**

(a) Finnish boreal forest

| Rate ($°C$ $min^{-1}$) | Temperature ($°C$) | Hold time (min) |
|---|---|---|
| | 35 | 5 |
| 6 | 180 | 5 |
| Total runtime: 34.0 min | | |

| Pressure (psi) | Average velocity (cm $sec^{-1}$) | Holdup time (min) |
|---|---|---|
| 44.25 (3.05 bar) | 44.02 | 2.27 |


(b) IAGOS-CARIBIC

| Rate ($°C$ $min^{-1}$) | Temperature ($°C$) | Hold time (min) |
|---|---|---|
| | 30 | 8 |
| 6 | 180 | 1 |
| Total runtime: 34.2 min | | |

| Pressure (psi) | Average velocity (cm $sec^{-1}$) | Holdup time (min) |
|---|---|---|
| 45.02 (3.10 bar) | 44.23 | 2.26 |

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
