# Peer review of "Atomic emission detector with gas chromatographic separation and cryogenic pre-concentration (CryoTrap-GC-AED) for atmospheric trace gas measurements"

_Atmospheric Measurement Techniques, 2020_

## Referee Comment (RC1) · Elliot Atlas (Referee) · 3 Aug 2020

The manuscript by Karu et al. describes a newly developed trace gas measurement system based on cryogenic enrichment of ambient air samples followed by gas chromatographic separation prior to compound detection and quantification based on atomic emission spectrometry. The novel aspect of the manuscript is the use of an atomic emission detector. As this detector has seen little use for routine trace gas analysis, this paper would be of interest to the atmospheric sciences community. However, the paper needs substantial modification before publication.

[Figure]

The authors wish to present the case that the features of atomic emission detection (equimolar atomic response independent of compound and specific atom detection) provide new capabilities for atmospheric chemical measurements. However, the authors undermine their case by demonstrating that, in fact, their combined analytical system produces a compound specific response, presumably related to the enrichment and pre-separation protocols. Thus, individual compound standards must be used for quantitation. Also, the examples of field measurements shown by the authors might have been done more easily with a GC/FID or GC/MS system. No advantage to the AED is demonstrated, nor is it compared to other techniques to show that it is at least equivalent to "standard" methods. It would have been valuable to see how the multi-atom capabilities of the AED could be used, for example, to identify unknowns by determining composition of a mixed halocarbon compound (or other hetero atom compound) from response factors of the component halogens (heteroatoms) and of carbon. Similarly, could the AED capability to measure oxygen be used to better identify oxygenated VOC that may co-elute in a complex air sample?

Other comments are listed below

L33. Even though "great care" was taken to minimize compound specific effects, such effects are later reported. Here the authors claim that a single RF could be applied to an unknown compound, but they don't indicate how they choose this RF, or if they use some average.

L41 The authors show a range of 161 – 211 nm for the range of the JAS AED III HR instrument. It seems from the literature of JAS that higher wavelengths are also accessible. Earlier AED models did have a much wider range. Can the authors clarify if this limited range is all that is available? Improved responses for some elements might be found at higher wavelengths. If the instrument is limited in wavelength range, perhaps the authors could comment on the advantages or disadvantages of a limited range versus an extended range instrument.

[Figure]

L47 The authors state that the performance of atomic emission lines for multiple compounds is discussed. However, no information is provided if the authors characterized different potential atomic emission lines for each atom. Why were these particular lines chosen? Or if different ratios of combustion gases were tested to evaluate the impact of plasma conditions on the results.

L56 Are the sample inlet lines temperature controlled? Is an internal standard used? Did you determine the maximum sample size possible before the most volatile measured compound breaks through the trap?

L64 Could you specify the sample and other flows used during sample collection/transfer?

L96 Perhaps also specify purity requirements for the H2 and O2.

L114 Could you specify what backflush and bakeout times were required based on the carryover experiments? What level of carryover is observed from ppb level to zero level samples?

L116 Only two standards are described, but it says that three were used. Please clarify. Later a NOAA ambient air calibration standard was mentioned. Is this number 3? If so, how does this compare with dilutions of the Riemer or NPL standards?

L131 Not really sure what linear up to 4 orders of magnitude means? Could you please provide a table of response factors determined for a range of different classes of compounds over the 4 orders of magnitude that were tested. Can you also do a similar table for the calibration levels used for the characterization experiments (e.g. 50 – 1000 pptv).

L137 I am curious how the instrument drifted under normal use. Is there a significant drift in system response over time?

L144 The LOD determination can be done in a variety of ways. In a system that is free of adsorption or other arifacts, the equation 1 might be a reasonable extrapolation to

determine detection limit. However, the system described does have some compound specific characteristics, so it would be helpful to verify the reported LODs with actual measurements near that LOD (e.g., within a factor of 3).

L152 It is not clear how the average RF that is calculated was actually used for quantitation.

L154 Were other carbon emission lines tested? If so, what were the different characteristics? Section 3.1 This section describes results that begs further investigation and discussion. The range of reported response for carbon in different compounds is very troubling. What is reported is not the equimolar response that is claimed for the system, and it suggests big problems somewhere. A factor of >2 between benzyl chloride and butyl nitrate seems to me to be a big problem that needs to be evaluated further. To me, the differences suggest issues of stability or standard drift, though system artifacts can't be ruled out. The observed differences could be checked with independent standards, or at least compare calculations of mixing ratios based on both carbon and heteroatoms in the molecule. Further, the large variability within each response factor also seems to be a problem (36% for butyl nitrate, 43% for benzyl chloride). It is not clear to me how these variations are useful for quantitative analysis. Overall, the response factor uncertainty (to 1 sigma) is quite variable, from reasonable few percent to much higher values depending on compounds.

The authors also note that the standard deviations from the Apel-Riemer standard is about 2 x that of the NPL standard. The authors speculate that this may result from changing discharge tubes, though one could also speculate that the dilution system used for the Apel-Riemer tests had some stability problems. If the problem is from discharge tubes, wouldn't this suggest a significant problem for routine application?

Paragraph beginning L178. This paragraph describes the crux of the problem with this paper and with the proposed method. The nominal advantage of the equimolar response of an AED is not found in this system, presumably because of sample preparation/separation issues. I don't think it lies with the AED, but this could be checked by additional testing. My suggestion would be to locate the source of the problems and determine what needs to be fixed to improve the accuracy and precisions of the whole instrument. As presented, the analytical system described has some basic flaws. It can, as the authors point out, be used if one calibrates each compound. But this is essentially no different from use of an FID, which is substantially less complicated. (I would be interested to know if the cryo-enrichment unit was tested using an FID detector, and how this might compare.) The average RF is precise to about 10% (only 1 SD), and this offers no advantage over other common analytical systems.

Section 3.2 Why is chlorine not included in this section? Was this not measured due to wavelength limitations? If available, please add Cl atom responses to this Table 2, and discuss in this section. It would have been great to also have Cl responses for the Apel Riemer standard to help diagnose the system. Also, I was trying to compare relative C and halogen, N, or S responses, but couldn't match all compounds between Table 1 and Table 2. For example, bromodichloromethane and acrylonitrile are in Table 2, but not listed in Table 1. Why?

L204. This sentence further explains the fundamental problem with the proposed method. "The different element count scales were non-related to each other." It seems to me that this negates the advantages of AED over other methods. Note for example CS2 and OCS. Ratio of C response between compounds (OCS/CS2): C ratio = 637/696 = 0.915; S ratio=342/476 = 0.718. How these differences are handled in practice needs clarification. It is not specified in the later examples, for example, if C or S emission lines are both used for the CS2 and OCS measurements, or one or the other. The authors might also wish to check on influence of CO2 on the response factors. I found a 1994 reference (Swan and Ivey, DOI: 10.1002/jhrc.1240171203) that discuss use of AED for ambient S compound analysis. Interestingly, their S ratio response between OCS and CS2 was 2932/1745 = 1.68, something quite different from that reported here (note: should add this reference to Introduction).

Section 3.3 The authors compare the advantages/disadvantages of the AED system to other common analytical systems. Given the problems outlined, the main advantage seemed to be the detection of compounds with low FID response, particularly COS. This is not particularly compelling, as other (simpler) options are available. The calculated detection limits are useful for ambient measurement of the NMHC, OCS, and perhaps several oxygenated species. LODs of halogenated species is too low for ambient measurement in most environments. The authors further argue that the multi-element capabilities of the system are useful for screening gas samples. I would agree, but the selective transmission of compounds through their system would make quantitative estimations problematic, especially for exotic compounds. It would have been much more interesting to see how this system might be used to measure volatile arsenic, selenium or silicon compounds, which are not routinely done by other methods. Furthermore, the issue of the deterioration of the plasma tube and potential effects on atomic emission would also seem to be a big disadvantage for routine analysis. Given that the instrument has been in use for several years, the authors may be able to comment on that in more detail.

Section 4 Case Studies. The case studies presented show that the AED system can produce data that appears reasonable. It would be important to help validate the analytical system if it was compared to currently validated techniques, or if the instrument was used in some sort of multi-lab comparison study. The forest study was mostly for terpenes and sulfur species. Were any new S species (or other heteroatom) species found in this environment? For the CARIBIC flights, one might get a better sense of sample to sample reproducibility if the data were presented in a graphical time series that also included flight altitude and perhaps ozone as a secondary tracer. Or, if the samples were measured in more than one lab, a comparison of the data for OCS would be of interest.

---

## Referee Comment (RC2) · Anonymous Referee #2 · 26 Aug 2020

Review of Karu, E., Li, M., Ernle, L., Brenninkmeijer, C. A. M., Lelieveld, J., and Williams, J.: Atomic emission detector with gas chromatographic separation and cryogenic pre-concentration (CryoTrap-GC-AED) for atmospheric trace gas measurements

This paper describes the development of a new GC analysis system for atmospheric VOC observations that incorporates a high-resolution AED, allowing low pptv range measurements over a wide concentration range. This is a very well written and interesting paper. The components of the system are well described, with clear diagrams, although there are a few comments below regarding technical details. The response of the instrument has been well characterised and the methodology, limits of detection and response discussed. I would like to see some further discussion on the variability of RF values for single compounds included. Two case studies of instrument deployment and laboratory characterisation of whole air samples show the applicability of the method, although it is not clear whether the new instrument has any advantages over exisiting methods for these analysis. I recommend publication in AMT subject to minor corrections and clarifications outline below.

Line 62: Is there any data to show that the adsorption to the Silonite-D layer to de-active surfaces is very low?

Figure 1: please say what part of the cycle this configuration is in, it took me a while to work out the flow regime in the valves. In the module 3, pre-concentration trap, does this contain packing or is it just a coated empty tube?

Line 98: Can you be sure 100 % conversion to elements in the AED? Could this also be part of the reason for compound RF variability?

Line 105: What is element installation?
Line 113: This sentence is not very clear. Please consider rewording.
Line 121: give details of the MFC

Line 147: Why are the RFs for heteroatom containing species given as a per C RF? Does Table 1 consider solely the C signal and not the other wavelengths? I can see when reading further that this is the case. Perhaps just include a sentence to describe that you will first look at C only and then consider the other elements separately.

Figure 1: Are all 84 species chromatographically separated in the C wavelength signal? This is essential if the C wavelength response is used. Also, how was the retention order determined? Was a GC-MS /retention indices used to determine the retention times prior to this detector being added?

Table 2: Is there a trend in RF or RF variability based on whether a compound is a hydrocarbon or hetero-VOC?

Figure 4: I would like to see more discussion of the reproducibility for each compound. In some cases the standard deviation seem high and there is a significant discrepancy between the two standards.

Line 196, Figure 2: It might be worth pointing out here that the high background is responsible for the higher LOD. The hetero-atom chromatograms show very useful selectivity and removal of the complex background signal.  This is not really discussed here but seems like it would be useful in ambient samples, although only if the reduction in LOD is not an issue.  Could you identify any other heteroatom containing species in the ambient samples, rather than just the targeted ones shown?  It would be good to see the chromatograms for the other elements for a real sample, rather than just a gas standard. I can see this is where the technique could have advantages over an FID. As presented, the case studies don't really provide exciting new data that couldn't be achieved with more standard instrumentation.

Line 236: perhaps direct reader that this information is about to be presented. When I first read this, I wondered why this information was given here.

Line 258: extra space in 1400
Line 301: why was the NPL standard only used at 50 mL? This seems odd considering the Apel-Riemer is at much higher concentrations
Line 325: should this say "marine OR volcanically"?

---

## Author Comment (AC1) · 29 Sep 2020

Author response to RC1 of Karu, E., Li, M., Ernle, L., Brenninkmeijer, C. A. M., Lelieveld, J., and Williams, J.: Atomic emission detector with gas chromatographic separation and cryogenic preconcentration (CryoTrap-GC-AED) for atmospheric trace gas measurements.

Here present an in-depth analysis of a potentially useful new instrument for atmospheric research, namely the CryoTrap-GC-AED system. The instrument was purchased to replace an ageing FID system, the aim being to simplify calibrations with the equimolar response ideally offered by this technique and to use the element specific signals with low associated noise to search for new species of atmospheric interest. In this paper we report that in fact the species measurable by this instrument do exhibit compound specific responses due to the sampling preparation steps, and that no exotic heteroatom containing species were discovered in the samples analyzed. Although apparently disappointing it is nonetheless important to report these findings. The initial manuscript submitted included the instrument description and several examples of how selected data was used to derive new atmospheric information (for example on the upper atmosphere budget of OCS). At the request of the editor this was split into this instrumental paper (here) and a separate data analysis paper. We agree with the reviewer that since no new species (containing of iodine, bromine, nitrogen, silicon, mercury, selenium and arsenic) were revealed in any of the sampled environments to date that a GC-MS targeting the compounds in the calibration gas would have served the same purpose. Although it should be noted the GC-MS could exhibit a much greater range of sensitivities to individual compounds. For transparency we state this point now clearly in the discussion as "L249: For the work presented here, a pre-concentrating GC-MS could have served as the same or better alternative for monitoring targeted species present in a calibration gas. Nevertheless, the AED system has advantages for searching exotic heteroatom molecules in the atmosphere, for the non-carbon elements which are in the detector wavelength range.". We take the view that it is important to document the instrument performance "warts and all".

Compound specific response – all compounds measured in the atmosphere were calibrated against a multi-component gas mixture, thus all the losses were accounted for.

Added to the manuscript at L190.

Identify unknowns by determining composition of a mixed halocarbon compound – Unfortunately, chlorine and fluorine do not land in the quantifiable 161 – 211 nm wavelength range (AED III HR has a restricted wavelength range, but all the wavelengths can be recorded simultaneously. AED II has a mechanically turning grating for measuring different wavelength groups in the range of 171 to 837 nm, thus nevertheless the simultaneous measurement of the whole range is not possible).

Added to the manuscript at L44.

Throughout my PhD, I was always on a search for new compounds in the group of iodine, bromine, nitrogen, silicon, mercury, selenium and arsenic but these elements were mostly under the quantifiable detection limit with the method developed, and in the regions measured.

Added to the manuscript at L251.

Might have been done more easily with a GC/FID or GC/MS – We agree that for the presented case these instruments would have been a better option, but as we were also searching for new heteroatom containing species in the samples, where AED detector has an advantage.

Added to the manuscript at L249.

AED capability to measure oxygen – not suitable for trace atmospheric measurements as the requirement is to use 10% methane 90% nitrogen reagent gas mixture. As this reagent gas produces a very high background signal it was not feasible to use it for trace gas measurements which was the aim of this project. This AED capability is suitable for oxygen measurements in oil and gas industry.

Added to the manuscript at L245.

*L33 Even though "great care" was taken to minimize compound specific effects, such effects are later reported. Here the authors claim that a single RF could be applied to an unknown compound, but they don't indicate how they choose this RF, or if they use some average.*

Compound specific RFs could be applied for a similar compound groups (pseudo-unknown compounds) within the uncertainty range. Even better approach would be to average similar compound group RFs to quantify the pseudo-unknown or a compound missing in the calibration standard.

Added to the manuscript at L35.

*L41 The authors show a range of 161 – 211 nm for the range of the JAS AEDIII HR instrument. It seems from the literature of JAS that higher wavelengths are also accessible. Earlier AED models did have a much wider range. Can the authors clarify if this limited range is all that is available? Improved responses for some elements might be found at higher wavelengths. If the instrument is limited in wavelength range, perhaps the authors could comment on the advantages or disadvantages of a limited range versus an extended range instrument.*

With the AEDIII HR wavelength range of only 161 – 211 nm can be measured simultaneously (light is focused in a horizontal plane onto two back-thinned CCDs). Due to a physical hardware gap between the two CCDs, there is a 7 nm gap in the wavelength spectra between 183 – 190 nm range). Therefore, e.g. chlorine and fluorine cannot be measured with the AEDIII HR version. For example, AEDII (previously developed by HP -> Agilent -> JAS GmbH) was able to measure in the range of 171 to 837 nm with a turning grating as mentioned above, therefore not all wavelengths simultaneously. Unfortunately, chlorine (sensitive at 479 nm) and fluorine (sensitive at 690 nm) are not in the measurement range of AEDIII HR. The advantage of the limited wavelength range is the simultaneous recording of the whole spectra with 0.01 nm step resolution.

Added to the manuscript at L44 and L108.

The newer development by the JAS company AED III Wide Range (WR) measures much wider range with movable optics. Thus, simultaneous recording of the whole spectral range is once again not possible (different elements must be recorded in different wavelength range groups). Therefore, also the resolution is lower and likely also the sensitivity (they do not state the sensitivity yet).

https://www.jas.de/en/products/jas/aed/aed-iii-wr-wide-range-spectrometer/

*L47 The authors state that the performance of atomic emission lines for multiple compounds is discussed. However, no information is provided if the authors characterized different potential atomic emission lines for each atom. Why were these particular lines chosen? Or if different ratios of combustion gases were tested to evaluate the impact of plasma conditions on the results.*

The most sensitive atomic emission lines in this setup and in the range of this detector were chosen. Iodine (163 nm), sulfur (181 nm), carbon (193 nm), bromine (163 nm) and nitrogen (174 nm). No secondary atomic emission lines for each element were used. The optimal ratio of hydrogen at 12 psi (0.83 bar) and oxygen (15 psi (1.03 bar) supply pressure was used throughout the measurements.

Added to the manuscript at L49 and L101.

*L56 Are the sample inlet lines temperature controlled? Is an internal standard used? Did you determine the maximum sample size possible before the most volatile measured compound breaks through the trap?*

All the sample inlet lines entering the CryoTrap instrument were at constant lab temperature and same length inlet lines were used for calibration standards and the sample lines, thus any losses here would be accounted for. No internal standard was used for these measurements. As the liquid nitrogen based trapping at -60°C to the main Tenax trap retains the analytes well, no breakthrough of the most volatile measured compounds was noticed with the maximum sample size of 1 800 mL tested.

Added to the manuscript at L120.

*L64 Could you specify the sample and other flows used during sample collection/transfer?*

Nominal sample trapping flow of 200 mL min$^{-1}$ was used and recorded. Helium flush flow rate was set to 100 mL min$^{-1}$. Trap module 1 to trap module 2 transfer flow of 10 mL min$^{-1}$ was used.

Added to the manuscript at L69, L71 and L72.

*L96 Perhaps also specify purity requirements for the $H_2$ and $O_2$.*

For the reagent gases extra ultra-high purity $H_2$ (EUHP purity > 99.99999% by a Parker Balston Hydrogen Generator, model H2-300, Parker Hannifin Corporation, USA) and ultra-high purity $O_2$ (UHP, purity 99.9999%, Westfalen, Germany) were used.

Added to the manuscript at L57.

*L114 Could you specify what backflush and bakeout times were required based on the carryover experiments? What level of carryover is observed from ppb level to zero level samples?*

With the 15 min bakeout step after the injection the modules 1 and 2 are heated to 150°C and 210°C respectively and back-flushed with He flow (75 ml min$^{-1}$), whereas the bulkhead heaters are kept at 150°C. After this bakeout no carryover was measured.

Added to the manuscript at L122.

*L116 Only two standards are described, but it says that three were used. Please clarify. Later a NOAA ambient air calibration standard was mentioned. Is this number 3? If so, how does this compare with dilutions of the Riemer or NPL standards?*

Sorry for the typo. Two standards were used for characterization experiments. For the IAGOS-CARIBIC sample measurement procedure, the National Oceanic and Atmospheric Administration (NOAA) ambient air calibration standard was used for calibrating carbonyl sulfide to the NOAA-scale.

*L131 Not really sure what linear up to 4 orders of magnitude means? Could you please provide a table of response factors determined for a range of different classes of compounds over the 4 orders of magnitude that were tested. Can you also do a similar table for the calibration levels used for the characterization experiments (e.g. 50 – 1000 pptv).*

With this sentence it is meant that linear calibration can be used at least from 1 pptv to 10 ppbv. It was a conservative measurement range, probably the linear range is even larger.

An example calibration curve for 6 species below:

[Figure]

L137 *I am curious how the instrument drifted under normal use. Is there a significant drift in system response over time?*

The most significant drift comes from discharge tube wear, but this was accounted for with frequent calibration standard measurement. Between different discharge tubes and the discharge fitting quality the background signal varied significantly.

L144 *The LOD determination can be done in a variety of ways. In a system that is free of adsorption or other artifacts, the equation 1 might be a reasonable extrapolation to determine detection limit. However, the system described does have some compound specific characteristics, so it would be helpful to verify the reported LODs with actual measurements near that LOD (e.g., within a factor of 3).*

For the LOD calculations the lowest practically achievable calibration points were used. As the compounds of interest in the unknown samples were almost always in the real calibration range, thus should not have a large effect.

L152 *It is not clear how the average RF that is calculated was actually used for quantitation.*

The average RF was used for unknown analyte determination derived from a linear calibration graph (area under curve [counts] vs. MR [pptv]).

L154 *Were other carbon emission lines tested? If so, what were the different characteristics? Section 3.1 This section describes results that begs further investigation and discussion. The range of reported response for carbon in different compounds is very troubling. What is reported is not the equimolar response that is claimed for the system, and it suggests big problems somewhere. A factor of >2 between benzyl chloride and butyl nitrate seems to me to be a big problem that needs to be evaluated further. To me, the differences suggest issues of stability or standard drift, though system artifacts can't be ruled out. The observed differences could be checked with independent standards, or at least compare calculations of mixing ratios based on both carbon and heteroatoms in the molecule. Further, the large variability within each response factor also seems to be a problem (36% for butyl nitrate, 43% for benzyl chloride). It is not clear to me how these variations are useful for quantitative analysis.*

*Overall, the response factor uncertainty (to 1 sigma) is quite variable, from reasonable few percent to much higher values depending on compounds. The authors also note that the standard deviations from the Apel-Riemer standard is about 2 x that of the NPL standard. The authors speculate that this may result from changing discharge tubes, though one could also speculate that the dilution system used for the Apel-Riemer tests had some stability problems. If the problem is from discharge tubes, wouldn't this suggest a significant problem for routine application?*

Only the carbon emission line at 193 nm as the most sensitive atomic emission line in the range was tested. The big RF differences could arise from different loss and trapping efficiencies in the whole analytical pathway as well as plasma atomization efficiency. Furthermore, the extra step of diluting the nominal 50 ppbv Apel-Riemer-2015 standard is an extra point for higher uncertainty compared to the direct trapping of the nominally 4 ppbv NPL-2017 calibration standard. Also, the stability differences within the calibration standards bottles could also play a role.

Added to the manuscript at L189 and L198.

*Paragraph beginning L178. This paragraph describes the crux of the problem with this paper and with the proposed method. The nominal advantage of the equimolar response of an AED is not found in this system, presumably because of sample preparation/separation issues. I don't think it lies with the AED, but this could be checked by additional testing. My suggestion would be to locate the source of the problems and determine what needs to be fixed to improve the accuracy and precisions of the whole instrument. As presented, the analytical system described has some basic flaws. It can, as the authors point out, be used if one calibrates each compound. But this is essentially no different from use of an FID, which is substantially less complicated. (I would be interested to know if the cryo-enrichment unit was tested using an FID detector, and how this might compare.) The average RF is precise to about 10% (only 1 SD), and this offers no advantage over other common analytical systems.*

Correct. Unfortunately, until now the CryoTrap system was not tested with an FID instrument. I agree, this would give significantly more insight to both, the CryoTrap preconcentration and thus indirectly also to the AED detector. Some comparison measurements were made between the CryoTrap-GC-AED and the GC-FID instrument where same sample air was analyzed. The results agreed within 10%. Nevertheless once again, as all the reported compounds were analyzed against calibration gas with the same parameters as unknown samples, thus all the losses were accounted for.

Added to the manuscript at L190.

*Section 3.2 Why is chlorine not included in this section? Was this not measured due to wavelength limitations? If available, please add Cl atom responses to this Table 2, and discuss in this section. It would have been great to also have Cl responses for the Apel Riemer standard to help diagnose the system. Also, I was trying to compare relative C and halogen, N, or S responses, but couldn't match all compounds between Table 1 and Table 2. For example, bromodichloromethane and acrylonitrile are in Table 2, but not listed in Table 1. Why?*

Yes, unfortunately chlorine is not in the range of this AED detector. The reason some compounds are missing from the carbon emission line determined response (Table 1) is mostly due to the co-eluting of the compounds on the C193 nm chromatogram, but clearly separated e.g. on the bromine chromatogram. In the case of the missing compounds, they will be added to Table 1.

*L204. This sentence further explains the fundamental problem with the proposed method. "The different element count scales were non-related to each other." It seems to me that this negates the advantages of AED over other methods. Note for example CS2 and OCS. Ratio of C response between compounds (OCS/CS2): C ratio = 637/696 = 0.915; S ratio=342/476 = 0.718. How these differences are*

*handled in practice needs clarification. It is not specified in the later examples, for example, if C or S emission lines are both used for the CS2 and OCS measurements, or one or the other. The authors might also wish to check on influence of CO2 on the response factors. I found a 1994 reference (Swan and Ivey, DOI: 10.1002/jhrc.1240171203) that discuss use of AED for ambient S compound analysis. Interestingly, their S ratio response between OCS and CS2 was 2932/1745 = 1.68, something quite different from that reported here (note: should add this reference to Introduction).*

All element emission lines were used independently and only one element's wavelength (the most sensitive for every specific compound) was used for calibration and quantification of the analytes of interest.

Added to the manuscript at L220.

Thank you for this reference, this is added to the manuscript.

Added to the manuscript at L27.

*Section 3.3 The authors compare the advantages/disadvantages of the AED system to other common analytical systems. Given the problems outlined, the main advantage seemed to be the detection of compounds with low FID response, particularly COS. This is not particularly compelling, as other (simpler) options are available. The calculated detection limits are useful for ambient measurement of the NMHC, OCS, and perhaps several oxygenated species. LODs of halogenated species is too low for ambient measurement in most environments. The authors further argue that the multielement capabilities of the system are useful for screening gas samples. I would agree, but the selective transmission of compounds through their system would make quantitative estimations problematic, especially for exotic compounds. It would have been much more interesting to see how this system might be used to measure volatile arsenic, selenium or silicon compounds, which are not routinely done by other methods. Furthermore, the issue of the deterioration of the plasma tube and potential effects on atomic emission would also seem to be a big disadvantage for routine analysis. Given that the instrument has been in use for several years, the authors may be able to comment on that in more detail.*

Agreed. During the measurement campaigns, we were always on the search for more exotic arsenic, selenium and silicon compounds but unluckily they were not detectable in the regions we measured and within the sensitivity of these elements for the system. CryoTrap-GC-MS is a better alternative for tasks monitoring targeted species. For example, the AED measurements could reveal exotic species from volcanic emission environments. This is now stated in the discussion section.

Added to the manuscript at L250.

*Section 4 Case Studies. The case studies presented show that the AED system can produce data that appears reasonable. It would be important to help validate the analytical system if it was compared to currently validated techniques, or if the instrument was used in some sort of multi-lab comparison study. The forest study was mostly for terpenes and sulfur species. Were any new S species (or other heteroatom) species found in this environment? For the CARIBIC flights, one might get a better sense of sample to sample reproducibility if the data were presented in a graphical time series that also included flight altitude and perhaps ozone as a secondary tracer. Or, if the samples were measured in more than one lab, a comparison of the data for OCS would be of interest.*

Until now, multi-lab comparison study has not been conducted but it would help to further characterize the instrument. In the boreal forest study, no new sulfur or other heteroatom species were detected with the method used. For the IAGOS-CARIBIC OCS, currently the CryoTrap-GC-AED is the only instrument measuring this species.

---

## Author Comment (AC2) · 30 Sep 2020

Author response to RC2 of Karu, E., Li, M., Ernle, L., Brenninkmeijer, C. A. M., Lelieveld, J., and Williams, J.: Atomic emission detector with gas chromatographic separation and cryogenic preconcentration (CryoTrap-GC-AED) for atmospheric trace gas measurements.

Thank you for your review and helpful feedback. The improvements as suggested have been carried out, helping to improve the manuscript. The responses to the review will follow below.

*Line 62: Is there any data to show that the adsorption to the Silonite-D layer to de-active surfaces is very low?*

Reference added:

Cardin, D.: Improving the Performance of Time Integrated Sampling of TO14 Compounds into Stainless Steel Canisters. Application Note: A-3725-02, 1999.

Added to the manuscript at L67.

*Figure 1: please say what part of the cycle this configuration is in, it took me a while to work out the flow regime in the valves. In the module 3, pre-concentration trap, does this contain packing or is it just a coated empty tube?*

The figure indicates the main sample trapping configuration through trap 1 and 2. The colors help to follow the flow path. The green color indicates the GC-AED column He flow and the blue color indicates the sample preconcentration flow pathway, starting from sample 1 inlet. Module 3 is just a part of 1/32" coated empty tube, no packing material inside.

To help clarify this we have inserted at line 371 the sentence "Blue color indicates the CryoTrap sample pre-concentration flow pathway through tarp 1 and 2, starting from sample 1 inlet position. Green color indicates the GC-AED He flow through to the column and leading to the He plasma.".

To include the information on module 3 a further sentence was also added at line 76.

*Line 98: Can you be sure 100 % conversion to elements in the AED? Could this also be part of the reason for compound RF variability?*

Good point, the conversion to atoms in the He plasma may not be 100%. This can also play a role in the compound RF variability.

RF variability may stem from less than 100% conversion of molecules to atoms in the plasma. Added to the manuscript at L189.

*Line 105: What is element installation?*

This is an automatic process of calibrating CCD diode numbers to wavelengths according to the calibration table.

Element installation is an automated process of calibrating CCD diode numbers to wavelengths.

Added to the manuscript at L110.

*Line 113: This sentence is not very clear. Please consider rewording.*

Done. "Furthermore, at the beginning of the experimental design the highest calibration standard MR level carryover and retention potentials were tested for all of the compounds of interest with the zero-air measurement directly after as the sample measurement."

Added to the manuscript at L119.

*Line 121: give details of the MFC*

Information added. "MFC, EL-Flow, Bronkhorst, the Netherlands"

Added to the manuscript at L129.

*Line 147: Why are the RFs for heteroatom containing species given as a per C RF? Does Table 1 consider solely the C signal and not the other wavelengths? I can see when reading further that this is the case. Perhaps just include a sentence to describe that you will first look at C only and then consider the other elements separately.*

Mistake corrected. The equation was written for the case of carbon emission line, but now written correctly in the general form. Table 1 is solely RF for per carbon atom, but Table 2 is RF per heteroatom.

Corrected in the manuscript at L156.

*Figure 1: Are all 84 species chromatographically separated in the C wavelength signal? This is essential if the C wavelength response is used. Also, how was the retention order determined? Was a GC-MS /retention indices used to determine the retention times prior to this detector being added?*

Most but not all of the 84 species are chromatographically separated in the carbon wavelength chromatogram. Co-eluting species that are detectable only in the carbon wavelength spectra are not reported. Retention times were determined with single species headspace "sniffing" tests and calibration gas mixtures.

*Table 2: Is there a trend in RF or RF variability based on whether a compound is a hydrocarbon or hetero-VOC?*

No clear trend was detected.

*Figure 4: I would like to see more discussion of the reproducibility for each compound. In some cases the standard deviation seem high and there is a significant discrepancy between the two standards.*

The discrepancy between two standards could have arisen from the extra dilution requirement of the 84 component nominal 50 ppbv Apel-Riemer-2015 calibration standard, which adds extra uncertainty. The nominal 4 ppbv NPL-2017 calibration standard needed no extra dilution.

The standard deviations of some compounds were higher than others, the Apel-Riemer standard being more variable. This was possibly the result of the extra dilution step required to dilute this standard to measurement level mixing ratios.

"In addition to the pre-concentration losses, the RF variability may also stem from less than 100% conversion of molecules to atoms in the plasma. As the per atom response factors determined to be compound specific, all the analytes of interest were always calibrated and quantified against a calibration standard, which was pre-concentrated and analyzed with exactly the same parameters as the samples. Therefore, all the losses and other effects in the analytical system leading to the variation in the RFs were accounted for."

"The discrepancy between the two calibration standard determined RFs seen in Table 1 and 2 could rise from the point that the compounds in the NPL-2017 calibration standard are at nominal 4 ppbv

and could be pre-concentrated directly, but the Apel-Riemer-2015 compounds are nominally at 50 ppbv, thus requiring extra dilution before the trapping. The dilution with zero-air introduces further uncertainty for the RFs determined from the Apel-Riemer-2015 calibration standard."

Added to the manuscript at L189 and L198.

*Line 196, Figure 2: It might be worth pointing out here that the high background is responsible for the higher LOD. The hetero-atom chromatograms show very useful selectivity and removal of the complex background signal. This is not really discussed here but seems like it would be useful in ambient samples, although only if the reduction in LOD is not an issue. Could you identify any other heteroatom containing species in the ambient samples, rather than just the targeted ones shown? It would be good to see the chromatograms for the other elements for a real sample, rather than just a gas standard. I can see this is where the technique could have advantages over an FID. As presented, the case studies don't really provide exciting new data that couldn't be achieved with more standard instrumentation.*

Great point. "Nevertheless, bromine emission line provides great selectivity, where the high background removes the complex background signal"

Added to the manuscript at L214.

Although we were always on the lookout for new heteroatom containing species in the ambient samples, we did not find any new compounds which were at a higher concentration than the LOD of the methods used.

"Unfortunately, no new exotic species (containing of iodine, bromine, nitrogen, silicon, mercury, selenium and arsenic) were revealed in any of the sampled environments to date."

Added to the manuscript at L252.

"For the work presented here, a pre-concentrating GC-MS could have served as the same or better alternative for monitoring targeted species present in a calibration gas. Nevertheless, the AED system has advantages for searching exotic heteroatom molecules in the atmosphere, for the non-carbon elements which are in the detector wavelength range. For example, to analyze volcanic emissions."

Added to the manuscript at L249.

*Line 236: perhaps direct reader that this information is about to be presented. When I first read this, I wondered why this information was given here.*

Done. "and will be presented in Chapter 4.1"

Added to the manuscript at L267.

*Line 258: extra space in 1400*

Done.

*Line 301: why was the NPL standard only used at 50 mL? This seems odd considering the Apel-Riemer is at much higher concentrations*

For routine measurements only one calibration MR was used for NPL-2017 standard (measured after every 5 unknown samples) for better statistics and time management.

*Line 325: should this say "marine OR volcanically"?*

You are correct, changed.

---

## Referee Report (RR1)

The authors have clarified a number of my questions regarding the operations of the current version of the AED III.  I appreciate that the authors wish to present an accurate characterization of their analytical system…"warts and all", and they do this to a certain extent.  However, there are mixed messages remaining in the paper that I think can be clarified.  The authors first sentence presents the AED as providing equimolar responses of elements (among other advantages).  However, they show that when the AED is combined with the cryo-GC-MS inlet that the responses are not equimolar, and the responses depend on many factors.  At several places in the manuscript, the authors will ascribe the variable response to either losses/artifacts in the inlet system or to inefficiencies of compound ionization in the AED plasma, or both.  So I don't know what the authors think they have learned about the AED, or about their inlet system.  Did they determine that there was a variable response to compounds in the AED plasma?  Did they determine there was an inlet problem?  Was it from the preconcentration step?  Was it from the GC column?  Did they determine that there were artifacts in the standard tanks or standard delivery system?  Various test could have been done to evaluate these potential issues, and more interpretation of the results is needed. I don't think the authors provided any particular insight into how their system operated.  Instead, the authors rely on doing what is necessary under these conditions: they use a multicomponent standard to account for all of the unknown factors that affect the system response to different compounds.  This is the practical solution, but not one which provides much useful information for the analytical chemist. It seems that the authors treat the Cryo-GC-AED system as a black box that can be calibrated to measure certain compounds within a relatively large standard deviation if one has appropriate standards.   To be fair, the manuscript reasonably describes the operation of the system which has been used for some field research.  It is useful to have this system documented in the literature.  It is a bit disappointing, though, since my own experience with an older version of the AED was much more inline with the advertised equimolar response, even with cryogenic or adsorbent preconcentration/GC separation prior to the detector.  This paper suggests otherwise and I think it is necessary for the authors to at least acknowledge that they have not been able to do the required testing to sort out reasons for the observed results.   I have seen a GC/AED respond to simple hydrocarbons from C3 – C8 with a constant response, which is also consistent with C response from various heteroatom compounds that were tested. It's troubling that the authors find something very different from their system.   My main suggestion for further revision is to more clearly discuss the "warts" and not gloss over the inconsistencies that were observed.

Below are some additional comments/suggestions:

L20     suggest "is reported to provide"…

L22     suggest  "may be attainable"

L28     change to "The first AED…

L35     here is an example that suggest inlet artifacts are minimized, but can't be true if AED is equimolar response as stated in L20.

L38     suggest change "pseudo-unknown" to "unidentified compound"

L40     note that previous AED systems also had sufficient sensitivity for ambient air, at least for C and for Cl in major CFCs and CH3Cl.

L46      actually MS can provide semiquantitative data on identified compounds without a specific standard

L55      probably should list the make/model of the tank regulators used in standard delivery system.

L139    I can see that the system is reasonably linear from the plots provided and from the good R^2. Still, I would like some uncertainty attached to that statement of linear over 4 orders of magnitude. I suggest a table in the supplement that shows the response factor for standards over those 4 orders of magnitude for at least a major selection of compounds, if not all of them.

L160-170        Please specify in the manuscript if the SD is one SD or 2 SD?  And if this description is the same throughout the text and tables.

It seems unfair to compare the overall C response factor in the Riemer mix, which contains many heteroatom compounds, with the NPL mix of all hydrocarbons. Could you do the comparison also of C response of just the hydrocarbons in the different mixes?

Given the difference between NPL and Riemer toluene response, can you say how you quantified toluene in ambient samples? One or the other, or average?

Do you think that the generally increasing variability of C response with increasing C number for hydrocarbons provides some insight into inlet vs AED issues?

L190    Here is an unsubstantiated claim that the AED response may not be 100% for different compounds.   At least note that you did not test this hypothesis, and that it is not consistent with manufacturer claims.

L198    but the observed difference for a few outliers can't be explained by dilution vs no dilution.   It seems to me that the NRL standard was compromised during the testing somehow. Looking at C response for hexane, benzene, toluene, ethylbenzene in the 2 standards suggests some issue with the NRL standard at least for toluene. Why not evaluate this difference rather than dismiss it?

| Standard | Hexane | Benzene | Toluene | EthylBenzene |
|---|---|---|---|---|
| Riemer | 643-+41 | 671-+53 | 643-+68 | 610-+92 |
| NRL | 701-+71 | 695-+65 | 913-+77 | 743-+101 |

L210    Not sure of the point of providing average S response if that is never used in practice.  Can you comment on the usefulness of a -+15% single standard deviation for trace gas analysis?

L230    Here is another example of the excessively optimistic presentation of AED characteristics.  I don't agree that a "great advantage" of an expensive, difficult to use AED is that it is more sensitive to OCS than the FID.   If the only alternative to OCS measurement was an FID, then this is an advantage, but other options for much better OCS measurements are available, including direct, high resolution cavity ring down direct measurements.

Field measurement performance        Given the authors comments on the plasma tube degradation issue, I think it is important to document the actual performance in continuous field operation. I would like to see a discussion of typical changes in C or S response vs time over the course of the experiments. This could provide additional useful comparison to the performance of other detectors.

L348.    Here again I don't know what the take-home message is. The authors state: "however, the application of the equimolarity feature of the detector is limited by pre-concentration and transmission losses."  Earlier the authors suggested the inlet was configured to minimize adsorption and losses, and that there could be <100% efficiency of ionization in the AED.  Here it seems that the authors do not question the "equimolarity feature" of the AED, but point to the inlet.   So what is the reader to believe?  And the authors might also want to consider testing several different standard tanks to evaluate various multicomponent mixes.

---

## Author Response (AR2)

Comments to the Editor:

Dear Marc,

We appreciate a lot for the helpful comments from both yourself and from the reviewers. We have tried our best to address the reviewers' comments. The major concerns from the reviewer are: what we have and have not learnt from the analytical system, and novelty of our study. As you have seen from the manuscript, we have a complete subsection 3.3 to discuss the advantages and disadvantages of the system, in a very open and transparent way. What we have learnt are: the non-equimolarity of the system which is in contrast to expectations, and linearity of its performance on VOC measurements. What we have not resolved is the underlying reason for the non-equimolar behavior and variable RFs in the two standard tanks. For our field measurements we resolved this problem pragmatically by calibrating all compounds individually to pressurized standards. The detail of these calibrations is well explained in the manuscript. Besides that, we have written in the manuscript that this behavior is a key disadvantage of the system. Furthermore, we have tried to explain the causes of this behavior and noted that further studies should investigate the reasons. The novelty of our study is that we present a newly developed analytical system which can provide specific information on selected elements. Although the system has some shortcomings (as with all analytical system), we have shown in our manuscript the potential of this system and how it may be operated. This should be interesting for AMT readers and for people interested in the new AED detector.

We have clearly seen that Elliot has suggested many tests to be done. We totally agree with that but please also consider the current COVID-19 lockdown measures in Germany, we are not able to conduct these experiments at the moment. However, we have stated specifically in the manuscript that further studies should be done to investigate the "unexplained" shortcomings of the system.

Best regards,

Einar Karu on behalf of all the authors

Comments to the Reviewer:

Dear Elliot,

We are grateful for your time and energy in providing further helpful comments on the manuscript. We have addressed the comments (in blue) carefully and our response are as follows (in black):

The authors have clarified a number of my questions regarding the operations of the current version of the AED III. I appreciate that the authors wish to present an accurate characterization of their analytical system…"warts and all", and they do this to a certain extent. However, there are mixed messages remaining in the paper that I think can be clarified. The authors first sentence presents the AED as providing equimolar responses of elements (among other advantages). However, they show that when the AED is combined with the cryo-GC-MS inlet that the responses are not equimolar, and the responses depend on many factors. At several places in the manuscript, the authors will ascribe the variable response to either losses/artifacts in the inlet system or to inefficiencies of compound ionization in the AED plasma, or both. So I don't know what the authors think they have learned about the AED, or about their inlet system. Did they determine that there was a variable response to compounds in the AED plasma? Did they determine there was an inlet problem? Was it from the preconcentration step? Was it from the GC column? Did they determine that there were artifacts in the standard tanks or standard delivery system? Various test could have been done to evaluate these potential issues, and more interpretation of the results is needed. I don't think the authors provided any particular insight into how their system operated. Instead, the authors rely on doing what is necessary under these conditions: they use a multicomponent standard to account for all of the unknown factors that affect the system response to different compounds. This is the practical solution, but not one which provides much useful information for the analytical chemist. It seems that the authors treat the Cryo-GC-AED system as a black box that can be calibrated to measure certain compounds within a relatively large standard deviation if one has appropriate standards. To be fair, the manuscript reasonably describes the operation of the system which has been used for some field research. It is useful to have this system documented in the literature. It is a bit disappointing, though, since my own experience with an older version of the AED was much more inline with the advertised equimolar response, even with cryogenic or adsorbent preconcentration/GC separation prior to the detector. This paper suggests otherwise and I think it is necessary for the authors to at least acknowledge that they have not been able to do the required testing to sort out reasons for the observed results. I have seen a GC/AED respond to simple hydrocarbons from C3 – C8 with a constant response, which is also consistent with C response from various heteroatom compounds that were tested. It's troubling that the authors find something very different from their system. My main suggestion for further revision is to more clearly discuss the "warts" and not gloss over the inconsistencies that were observed.

We are glad that the reviewer has noticed our best effort to present our analytical system in an accurate way.  Regarding "equimolar" response: In the manuscript it we now write that atomic spectrometric analysis has the potential advantage of equimolarity and that this will be tested in this paper for the new AED detector in combination with a typical GC and preconcentration set-up. The producing company claims the advantage of equimolarity for the AED version III (with liquid injection GC) and this is also reported from previous publications with older AED detector versions. In the previous manuscript it was not written that CryoTrap-GC-AED is equimolar, rather the opposite was determined for this setup and also written in the previous manuscript (see lines 186 and 348 in the 30.09.2020 manuscript version).. Discovering that the system is not equimolar when combined with

commercially available GC and cryoconcentrating devices, is important to those considering the AED system. We agree with the reviewer that further tests could been done to investigate the reasons on the variable response. Unfortunately due to the current COVID pandemic lockdown and lab regulations, we are unable to make these tests done in the near future. We now follow the suggestion from the reviewer to acknowledge that we have not resolved the reasons for the observed unequimolarity. We have added this sentence in the Conclusions section: "Further tests are needed to investigate the causes of variable responses to the compounds, and the underlying reasons for the unequimolar behavior of the system."

L20 suggest "is reported to provide"…
Changed to suggested.
L22 suggest "may be attainable"
Changed to suggested.
L28 change to "The first AED…
Changed to suggested.
L35 here is an example that suggest inlet artifacts are minimized, but can't be true if AED is equimolar response as stated in L20.
Sentence removed. Great care was taken to optimize the system. As short as possible transferlines made out of inert materials for the compounds of interest were used, system was often checked to be leak-tight, purest gases obtainable were used with extra scrubbers etc. Unfortunately the non-equimolarity remained using commercially available GC and cryofocussing systems and taking the aforementioned steps in the operation of the system.
L38 suggest change "pseudo-unknown" to "unidentified compound"
Changed to suggested.
L40 note that previous AED systems also had sufficient sensitivity for ambient air, at least for C and for Cl in major CFCs and $CH_3Cl$.
Sorry for the unclarity.Now changed to the following "Recently, further technical developments in the AED have led to improvements in sensitivity and furthermore, the whole AED III HR detector range is simultaneously measurable, making such systems potentially more attractive to atmospheric scientists."
L46 actually MS can provide semiquantitative data on identified compounds without a specific standard
Good point. Changed to the following to make it clearer: "Furthermore, the newly developed system can in principle provide unidentified compound quantification on a semiquantitative basis using identified compounds in a chromatogram but without a specific standard when a broad range of compound specific RFs are used for the determination of the single element RF. Such similar semiquantitative approaches can be also used with other analytical techniques, such as FID and MS."
L55 probably should list the make/model of the tank regulators used in standard delivery system.
Manuscript changed as following: "For the calibration gases and synthetic air as dilution gas Messer and Air Liquide gas regulators were used. For the certified ambient air calibration gas a pressure regulator completely made out of high-purity steel was used (Parker-Hannifin Veriflo 959, USA)."
L139 I can see that the system is reasonably linear from the plots provided and from the good R^2. Still, I would like some uncertainty attached to that statement of linear over 4 orders of magnitude. I suggest a table in the supplement that shows the response factor for standards over those 4 orders of magnitude for at least a major selection of compounds, if not all of them.
L160-170 Please specify in the manuscript if the SD is one SD or 2 SD? And if this description is the same throughout the text and tables.
If not otherwise stated, always 1σ SD was used. For example, in table 1 and 2 3σ level SD was used for LOD, where it is also stated. For clarity we now state this explicitly in table 1 and table 2 "± standard deviation on 1σ level"

It seems unfair to compare the overall C response factor in the Riemer mix, which contains many heteroatom compounds, with the NPL mix of all hydrocarbons. Could you do the comparison also of C response of just the hydrocarbons in the different mixes?

The 1:1 comparison of the compounds that were analyzed from both standards is shown in Fig. 3. We now additionally give the correlation for the simple alkanes.

Given the difference between NPL and Riemer toluene response, can you say how you quantified toluene in ambient samples? One or the other, or average?

Toluene measured by the AED system was not reported for these measurement campaigns due to the differing responses.

Do you think that the generally increasing variability of C response with increasing C number for hydrocarbons provides some insight into inlet vs AED issues?

A general increasing variability in carbon response is visible with increasing carbon atom number, which can be seen in Fig. 4. This is rather an indication that the variability increases due to CryoTrap trapping or inlet system.

L190 Here is an unsubstantiated claim that the AED response may not be 100% for different compounds. At least note that you did not test this hypothesis, and that it is not consistent with manufacturer claims.

Totally agreed, added the following sentence "The latter is a hypothesis and has not been tested.".

L198 but the observed difference for a few outliers can't be explained by dilution vs no dilution. It seems to me that the NRL standard was compromised during the testing somehow. Looking at C response for hexane, benzene, toluene, ethylbenzene in the 2 standards suggests some issue with the NRL standard at least for toluene. Why not evaluate this difference rather than dismiss it?

| Standard | Hexane | Benzene | Toluene | EthylBenzene |
|---|---|---|---|---|
| Riemer | 643-+41 | 671-+53 | 643-+68 | 610-+92 |
| NRL | 701-+71 | 695-+65 | 913-+77 | 743-+101 |

One option would be wall losses. From the comparison of response factors between the NPL and Apel-Riemer standards toluene appears to have been compromised somehow in the NPL standard. Wall losses are a possible reason for an anomalously high RF. Changed to "The dilution with zero-air could introduce further uncertainty, as well as other unknown factors (e.g. wall losses) may also cause this discrepency."

L210 Not sure of the point of providing average S response if that is never used in practice. Can you comment on the usefulness of a -+15% single standard deviation for trace gas analysis?

In the manuscript the CryoTrap-GC-AED performance was was described as well as how the experiments were done and what where the results. It was demonstrated in which range the semiquantitative measurements could be with this system. Nevertheless, a direct calibration was always used for field campaigns, these numbers (-+15% single standard deviation ) are interesting for comparison. A ±15% single standard deviation is not sufficiently accurate for atmospheric measurements slowly changing long-lived gases such as CFCs but helpful when searching for new emissions.

L230 Here is another example of the excessively optimistic presentation of AED characteristics. I don't agree that a "great advantage" of an expensive, difficult to use AED is that it is more sensitive to OCS than the FID. If the only alternative to OCS measurement was an FID, then this is an advantage, but other options for much better OCS measurements are available, including direct, high resolution cavity ring down direct measurements.

Agreed! Sentence removed completely. Added an alternative sentence instead: "The advantage of the AED based system is the highly selective element specific information obtainable from the emission wavelength data, which can be useful in helping to determine unknowns in a complex matrix."

Field measurement performance Given the authors comments on the plasma tube degradation issue, I think it is important to document the actual performance in continuous field operation. I would like to see a discussion of typical changes in C or S response vs time over the course of the experiments. This could provide additional useful comparison to the performance of other detectors.

During the measurements a calibration point was measured after every 5 samples to minimize the uncertainty risen from the response change.

L348. Here again I don't know what the take-home message is. The authors state: "however, the application of the equimolarity feature of the detector is limited by pre-concentration and transmission losses." Earlier the authors suggested the inlet was configured to minimize adsorption and losses, and that there could be <100% efficiency of ionization in the AED. Here it seems that the authors do not question the "equimolarity feature" of the AED, but point to the inlet. So what is the reader to believe? And the authors might also want to consider testing several different standard tanks to evaluate various multicomponent mixes.

Agreed. Sentence removed. Now added: "... and provides accurate element specific information." Furthermore, the following sentence was inserted: "The main reason for the higher variability in the RFs seems to arise from the CryoTrap preconcentration and inlet system.".

Hope these answers and the corrections to the manuscript help to make this paper more easily readable and demonstrates the work done and which results were obtained.

Best regards,

Einar Karu on behalf of all the authors

[revised manuscript text omitted]